# Chromatin accessibility and gene expression during adipocyte differentiation identify context-dependent effects at cardiometabolic GWAS loci

**Hannah J. Perrin**[1,&], **Kevin W. Currin**[1,&], **Swarooparani Vadlamudi**[1], **Gautam K. Pandey**[1], **Kenneth K. Ng**[1], **Martin Wabitsch**[2], **Markku Laakso**[3], **Michael I. Love**[1,4], **Karen L. Mohlke**[1]*

**1** Department of Genetics, University of North Carolina, Chapel Hill, North Carolina, United States of America, **2** Department of Pediatrics and Adolescent Medicine, Ulm University Hospital, Ulm, Germany, **3** Department of Medicine, University of Eastern Finland and Kuopio University Hospital, Kuopio, Finland, **4** Department of Biostatistics, University of North Carolina, Chapel Hill, North Carolina, United States of America

& These authors contributed equally to this work.

* mohlke@med.unc.edu

**Data Availability Statement:** All ATAC-seq and RNA-seq data are available from the GEO database (accession number GSE178796).

## Abstract

Chromatin accessibility and gene expression in relevant cell contexts can guide identification of regulatory elements and mechanisms at genome-wide association study (GWAS) loci. To identify regulatory elements that display differential activity across adipocyte differentiation, we performed ATAC-seq and RNA-seq in a human cell model of preadipocytes and adipocytes at days 4 and 14 of differentiation. For comparison, we created a consensus map of ATAC-seq peaks in 11 human subcutaneous adipose tissue samples. We identified 58,387 context-dependent chromatin accessibility peaks and 3,090 context-dependent genes between all timepoint comparisons (log2 fold change>1, FDR<5%) with 15,919 adipocyte- and 18,244 preadipocyte-dependent peaks. Adipocyte-dependent peaks showed increased overlap (60.1%) with Roadmap Epigenomics adipocyte nuclei enhancers compared to preadipocyte-dependent peaks (11.5%). We linked context-dependent peaks to genes based on adipocyte promoter capture Hi-C data, overlap with adipose eQTL variants, and context-dependent gene expression. Of 16,167 context-dependent peaks linked to a gene, 5,145 were linked by two or more strategies to 1,670 genes. Among GWAS loci for cardiometabolic traits, adipocyte-dependent peaks, but not preadipocyte-dependent peaks, showed significant enrichment (LD score regression P<0.005) for waist-to-hip ratio and modest enrichment (P < 0.05) for HDL-cholesterol. We identified 659 peaks linked to 503 genes by two or more approaches and overlapping a GWAS signal, suggesting a regulatory mechanism at these loci. To identify variants that may alter chromatin accessibility between timepoints, we identified 582 variants in 454 context-dependent peaks that demonstrated allelic imbalance in accessibility (FDR<5%), of which 55 peaks also overlapped GWAS variants. At one GWAS locus for palmitoleic acid, rs603424 was located in an adipocyte-dependent peak linked to SCD and exhibited allelic differences in transcriptional activity in

**Funding:** This study was supported by National Institutes of Health grants T32HL069768 (H.J.P.), F31HL146121 (K.W.C.), R25GM055336 (K.W.C.), T32GM67553 (K.W.C.), R01HG009937 (M.I.L.), U01KD105561 (K.L.M.), R01DK093757 (K.L.M.), and UM1DK126185 (K.L.M.); Academy of Finland grants 77299, 124243, 141226 (M.L.); Finnish Diabetes Foundation (M.L.); Finnish Heart Foundation (M.L.); and the Commission of the European Community grant HEALTH-F2- 2007-201681 (M.L.). The funders had no role in study design, data collection and analysis, decision to publish, or preparation of the manuscript.

**Competing interests:** The authors have declared that no competing interests exist.

adipocytes (P = 0.003) but not preadipocytes (P = 0.09). These results demonstrate that context-dependent peaks and genes can guide discovery of regulatory variants at GWAS loci and aid identification of regulatory mechanisms.

## Author summary

Cardiovascular and metabolic diseases are widespread, and an increased understanding of genetic mechanisms behind these diseases could improve treatment. Chromatin accessibility and gene expression in relevant cell contexts can guide identification of regulatory elements and genetic mechanisms for disease traits. A relevant context for cardiovascular and metabolic disease traits is adipocyte differentiation. To identify regulatory elements and genes that display differences in activity during adipocyte differentiation, we profiled chromatin accessibility and gene expression in a human cell model of preadipocytes and adipocytes. We identified chromatin regions that change accessibility during differentiation and predicted genes they may affect. We also linked these chromatin regions to genetic variants associated with risk of disease. At one genomic region linked to fatty acids, a chromatin region more accessible in adipocytes linked to a fatty acid synthesis gene and exhibited allelic differences in transcriptional activity in adipocytes but not preadipocytes. These results demonstrate that chromatin regions and genes that change during cell context can guide discovery of regulatory variants and aid identification of disease mechanisms.

## Introduction

Genome-wide association studies (GWAS) have identified thousands of loci associated with cardiometabolic traits, yet most mechanisms remain unclear due to unknown functional variants, genes, cell types, and relevant contexts, especially at the large number of noncoding loci [1]. Noncoding loci can regulate gene expression in cell-type and context-dependent manners [2]. Some GWAS loci colocalize with gene expression quantitative trait loci (eQTL) in trait-relevant tissues [3–8], although other GWAS loci colocalize with eQTL found only in one context, such as stimulated, but not naïve, immune cells [9]. Therefore, mapping transcriptional regulatory elements and gene expression in disease-relevant contexts can be used to characterize molecular mechanisms of GWAS loci. Enhancers and other regulatory elements can be detected by identifying regions of chromatin accessibility [10] using sequencing methods such as the Assay for Transposase Accessible Chromatin (ATAC-seq) [11]. Chromatin accessibility in cardiometabolic-relevant cell types and contexts can be integrated with GWAS and eQTL data to identify regulatory elements and variants that alter gene expression to affect cardiometabolic traits.

Adipose tissue influences cardiometabolic traits such as body fat distribution, insulin sensitivity, blood cholesterol levels, and inflammation through its roles in lipid storage and hormone secretion [12,13]. Hundreds of GWAS loci for cardiometabolic traits are colocalized with eQTL in adipose tissue [3–5], and variants at GWAS loci for some cardiometabolic traits, such as waist-to-hip ratio adjusted for body mass index (BMI) and high-density lipoprotein (HDL) cholesterol, are overrepresented in transcriptional regulatory elements in adipose tissue [14,15]. At a subset of colocalized GWAS-eQTL signals, adipose tissue gene expression may mediate the effect of the genetic variant on GWAS traits [5]. Adipose is a heterogeneous tissue

that contains multiple cell types, including adipocytes, preadipocytes, immune cells, and vascular cells [16]. Adipose tissue stores lipids through either hyperplasia, during which preadipocytes differentiate into mature adipocytes to store excess energy, or hypertrophy, during which existing adipocytes expand to store excess energy [17]. Thus, identifying variants with regulatory effects at specific stages of adipocyte differentiation may uncover additional mechanisms at GWAS loci for cardiometabolic traits.

Genetic and environmental variation between individuals can contribute to differences in chromatin accessibility [18]. Chromatin accessibility maps generated from multiple individuals can capture accessible regions that reflect genetic effects and diverse environmental contexts. Existing human adipose tissue chromatin accessibility maps are comprised of data from one to six individuals and differ by tissue donor characteristics (e.g. BMI, age, sex), adipose depot, tissue extraction site, and storage conditions [14,18,19]. Given the cell-type heterogeneity of tissue samples, it is also useful to characterize the component cell types in controlled environments. Chromatin accessibility during adipogenesis has been studied in models such as mouse 3T3-L1 cells [20], however additional studies in human models could improve interpretation of human non-coding genetic variants. Simpson-Golabi-Behmel Syndrome (SGBS) cells are a well-characterized diploid preadipocyte cell model that can be differentiated into mature adipocytes and is useful for studying adipocyte differentiation [21,22].

In this study, we identified differences in chromatin accessibility and gene expression between preadipocytes, immature adipocytes, and mature adipocytes in SGBS cells. In addition, we generated a consensus map of subcutaneous adipose tissue chromatin accessibility using 11 samples obtained from METabolic Syndrome in Men (METSIM) participants [23]. We used three methods to link differentially accessible regulatory elements to candidate genes and identified variants at cardiometabolic GWAS loci that resided in elements more accessible in preadipocytes or adipocytes. Finally, we identified variants at the *SCD* and *EYA2* loci that showed context-dependent and/or allelic effects on transcriptional activity, detecting potential mechanisms by which specific variants alter gene expression to affect cardiometabolic traits.

## Results

### Changes in chromatin accessibility across adipocyte differentiation timepoints identify context-dependent regulatory elements

We profiled chromatin accessibility during adipocyte differentiation with ATAC-seq in SGBS cells [11,24]. We analyzed a final set of ten replicates of preadipocytes (D0), ten replicates of immature adipocytes differentiated for four days (D4), and five replicates of mature adipocytes differentiated for fourteen days (D14) (Fig 1A and S1 Table). Our libraries had ~33–156 million filtered reads each, and showed high quality, with an average transcription start site (TSS) enrichment of 6.8, and an average fraction of reads in peaks (FRiP) of 48.5%. To test for differences in chromatin accessibility between timepoints, we generated a set of 147,587 accessible chromatin regions (ATAC peaks) at any time point (S2 Table) by merging the top 100,000 consensus peaks for each time point (ranked by median peak p-value across replicates, see Methods). Principal component analysis (PCA) showed that replicates clustered by differentiation timepoint, with preadipocytes and adipocytes separated by the first principal component, which explained 74% of the variance (S1 Fig and S1 Table).

To predict regulatory elements involved in adipocyte differentiation, we identified differentially accessible peaks, hereafter called 'context-dependent peaks', between each pairwise comparison of the three timepoints ($\log_2$ fold change (LFC)>1; false discovery rate (FDR)<5%; S3 and S4 Tables and S2 Fig). Based on the 10,000 context-dependent peaks with the most significant difference in any comparison, a heatmap showed that replicates clustered by timepoint

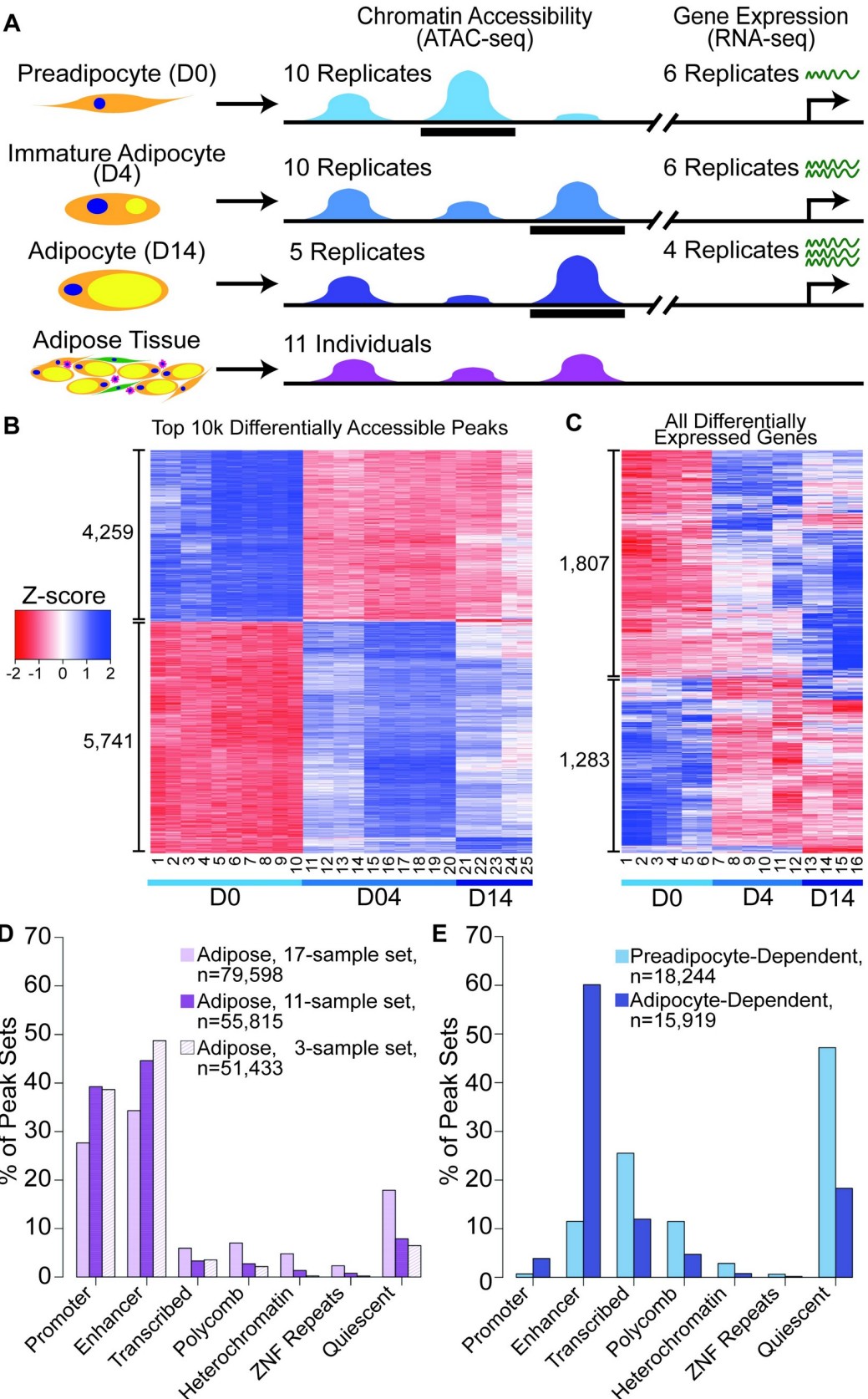

**Fig 1. Genome-wide profiles of chromatin accessibility and gene expression at three timepoints of adipocyte differentiation.** (A) Schematic of experimental design. SGBS cells were harvested as preadipocytes (D0), immature adipocytes (D4), and adipocytes (D14). Chromatin accessibility (blue) and gene expression (green) profiles were generated on replicates from each timepoint. Context-dependent peaks are shown as black bars. Chromatin accessibility profiles also were generated from subcutaneous adipose tissue (purple) of 17 individuals and an optimized consensus CA map was developed from a subset of 11 individuals. (B) Heatmap of the top 10,000 context-dependent peaks (from S4 Table) colored by z-score. Library numbers correspond to quality metrics in S1 Table. (C) Heatmap of expression level of all 3,090 context-dependent genes (from S9 Table) colored by z-score. Library numbers correspond to quality metrics in S8 Table. (D-E) Values in S9 Table. (D) Adipose peak overlap with chromatin states of Roadmap Epigenomics Project adipose nuclei for three sets of adipose consensus peaks. (E) Preadipocyte- and adipocyte-dependent peak overlap with chromatin states of Roadmap adipose nuclei.

(Fig 1B). Most (86%) of the changes in chromatin accessibility between D0 and D14 were observed by D4 (S2 Fig), and only 233 peaks were specifically more accessible in mature adipocytes (D14>D0 and D14>D4; S2 Fig), suggesting that chromatin accessibility changes early after the initiation of differentiation and remains largely stable between D4 and D14. To characterize the major differences, we identified context-dependent peaks more accessible in preadipocytes in both comparisons (D0>D4 and D0>D14; 18,244 peaks, S2 Fig), hereafter called 'preadipocyte-dependent peaks', and context-dependent peaks more accessible in immature and mature adipocytes (D4>D0 and D14>D0; 15,919 peaks, S2 Fig), hereafter called 'adipocyte-dependent peaks'. In analyses described below, we used the preadipocyte-dependent and adipocyte-dependent peaks for enrichment analyses and general comparisons between preadipocytes and adipocytes, and we used context-dependent peaks from all pairwise comparisons to identify regulatory elements linked to genes and GWAS loci.

We evaluated the relevance of context-dependent peaks for biological processes and transcription factors known to be involved in adipocyte differentiation. Preadipocyte-dependent peaks were enriched ($P < 1 \times 10^{-10}$) near genes associated with roles in several cell cycle processes, including positive regulation of DNA replication (S5 Table). Mature adipocyte-dependent peaks were enriched near genes with roles in cardiovascular development (S5 Table). Adipocyte-dependent peaks were enriched near genes with roles in several metabolic processes, including response to insulin, regulation of fatty acid oxidation, and intracellular lipid transport (S5 Table). In addition, preadipocyte-dependent peaks were enriched ($P < 1 \times 10^{-5}$) for transcription factor motifs for TEAD and GATA, which inhibit adipocyte differentiation [25,26], while adipocyte-dependent peaks were enriched for motifs of transcription factors that promote adipogenesis, such as CEBP, PPAR, and LXR [17,27] and transcription factors involved in glucose metabolism such as GRE [28] (S6 Table). Thus, adipocyte- and preadipocyte-dependent peaks are found near genes and contain transcription factor motifs relevant to their cell contexts, increasing confidence that these peaks capture relevant biology. Although genomic proximity between regulatory elements and genes is a strong predictor of a regulatory relationship [29], regulatory elements may not always affect the nearest genes.

To compare these SGBS peaks to adipose tissue peaks, we expanded our previous set of adipose tissue ATAC-seq profiles [14] from 3 to 17 samples that fulfilled sequencing quality thresholds (Methods, S7 Table and S3 Fig). In the 17 tissue samples, we identified 79,598 consensus adipose tissue peaks present in three or more samples. After removing 6 outlier samples identified using PCA, overlap with adipose regulatory elements, and other factors, we also identified 51,855 consensus adipose tissue peaks using 11 adipose tissue samples. The 11-sample peak set had a higher percentage of peaks within the Roadmap Epigenomics Project [30] adipose nuclei enhancers and promoters (45% enhancer, 39% promoter) compared to the 17-sample peak set (34% enhancer, 28% promoter), and a similar percentage compared to our previous 3-sample peak set [14] (49% enhancer, 39% promoter) (Figs 1D and S3 and

S8 Table). We proceeded with the 11-sample consensus adipose peak set for further analyses because it provides higher consistency with Roadmap adipose enhancers and promoters relative to the 17-sample set and may capture more genetic and environmental variation in chromatin accessibility than the 3-sample set.

To determine if context-dependent SGBS peaks marked previously annotated adipose regulatory elements, we compared the SGBS peaks to Roadmap Epigenomics Project adipose nuclei chromatin states [30] and to the 11-sample adipose tissue peaks. A higher percentage of adipocyte-dependent peaks were found within Roadmap adipose nuclei enhancers and promoters (60% enhancer, 3.9% promoter) compared to preadipocyte-dependent peaks (12% enhancer, 0.73% promoter) (Fig 1E and S8 Table). Similarly, 36% of adipocyte-dependent peaks overlapped (shared at least 1 base) adipose tissue peaks, while only 1.8% of preadipocyte-dependent peaks overlapped adipose tissue peaks, consistent with adipose tissue containing more adipocytes than preadipocytes [16,31]. Peaks found in SGBS and adipose tissue may have more relevance to adipose biology than peaks found in SGBS cells alone. These results show that our adipocyte-dependent and consensus adipose tissue peaks demonstrate strong similarity with existing adipocyte promoters and enhancers.

## Changes in gene expression across adipocyte differentiation

We generated RNA-seq data from six replicates of SGBS preadipocytes (D0), six replicates of immature adipocytes differentiated for four days (D4), and four replicates of mature adipocytes differentiated for fourteen days (D14) (Fig 1A). We generated ~36–56 million filtered reads overlapping transcripts per replicate (S9 Table) and identified 18,299 expressed genes (median normalized count $> = 1$ across libraries); S2 Table). PCA showed that replicates clustered by differentiation timepoint, with preadipocytes and adipocytes separated by the first principal component, which explained 54% of the variance (S4 Fig and S9 Table).

To identify changes in gene expression during adipocyte differentiation, we identified genes differentially expressed between each pairwise comparison of the three timepoints (LFC>1; FDR<5%; S3 and S10 Tables). A heatmap of these 'context-dependent genes' showed that replicates clustered by timepoint (Fig 1C). In addition, we identified context-dependent genes that were observed in multiple timepoint comparisons (S5 Fig and S10 Table). In contrast to context-dependent chromatin accessibility, for which 86% of changes between D0 and D14 were observed already by D4, only 1,282 of 2,107 (61%) context-dependent genes between D0 and D14 were observed already by D4 (S5 Fig). Although further analysis is needed, this result is consistent with previous studies that identified changes in chromatin accessibility that occurred earlier during adipocyte differentiation and remained more stable than changes in gene expression [20,32].

We tested context-dependent genes for enrichment of biological processes known to be involved in adipocyte differentiation. Genes expressed more strongly in preadipocytes than adipocytes were enriched ($P<1x10^{-10}$) for several cell cycle processes including cell cycle regulation and nuclear division (S11 Table). Genes expressed more strongly in adipocytes than preadipocytes showed enrichment ($P<1x10^{-10}$) for several differentiation and metabolic processes such as response to insulin, glucose homeostasis, fatty acid metabolic processes, and lipid biosynthetic processes (S11 Table). We also identified context-dependent transcription factors whose binding motifs were enriched in context-dependent peaks, including preadipocyte-dependent *GATA* family members that had motifs enriched in preadipocyte-dependent peaks, and the adipocyte-dependent gene *PPARG* whose motifs were enriched in adipocyte-dependent peaks (S6 and S10 Tables). Adipocyte-dependent genes also included known adipocyte-

dependent genes such as *ADIPOQ* [33]. These results indicate that the context-dependent genes have functions relevant to the corresponding cell types.

## Three approaches to link genes to context-dependent peaks

Linking context-dependent peaks to genes remains challenging because most peaks are located in non-coding regions with multiple genes nearby. Approaches to predict genes affected by a peak have varied sensitivity and specificity [2], thus we used three approaches to identify additional genes and to gain confidence in genes identified by more than one method. The three approaches used to link context-dependent peaks to genes were: overlap with adipocyte promoter capture Hi-C [34,35], overlap with adipose eQTL variants [5], and context-dependent expression of genes linked by either of the first two approaches (Fig 2A–2C).

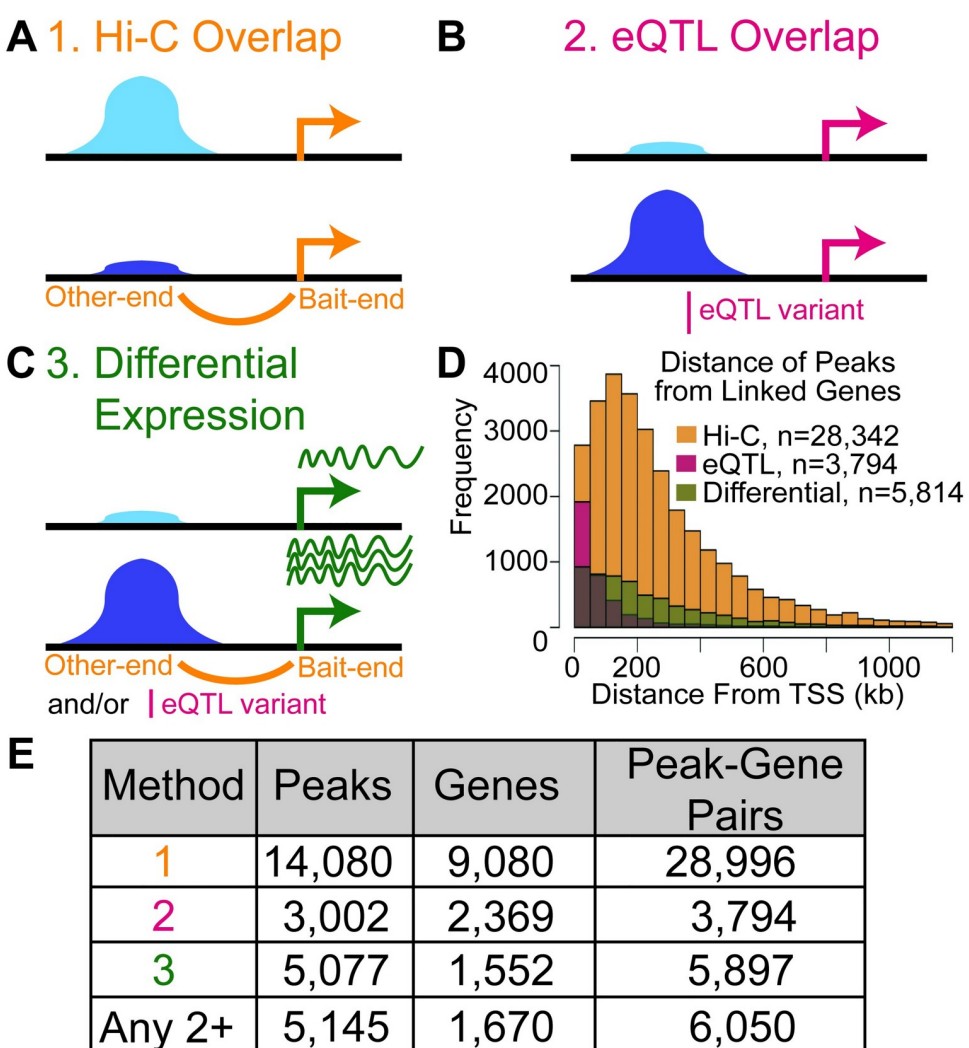

**Fig 2. Linking context-dependent chromatin accessibility to candidate genes.** (A-C) Schematic of three approaches to link peaks to genes. Day 0 (light blue) and day 14 (dark blue) context-dependent peaks are represented. (A) Context-dependent peaks that overlap elements connected to gene promoters using adipocyte promoter capture Hi-C (orange). (B) Context-dependent peaks that overlap adipose gene eQTL variants ($r^2$>.8 with lead, red). (C) Context-dependent peaks linked to a gene through Hi-C or eQTL for which the linked gene was also differentially expressed between any timepoints (green). (D) Histogram of distances from edges of peaks to the transcription start site of a linked gene within 1.2 Mb. Values in S12 Table. (E) Numbers of context-dependent peaks linked to genes by each method and by two or more methods. Values summarize full results in S12 Table.

In the first approach, we identified context-dependent peaks that overlapped adipocyte promoter capture Hi-C regions [34,35] (overlap> = 1 base pair, Fig 2A and S12 Table). We identified 14,080 peaks linked to 9,080 genes (28,696 peak-gene pairs). We investigated the extent to which increasing the overlap threshold between peaks and Hi-C fragments would change our results. Of the 14,594 peak-Hi-C fragment overlapping pairs (some peaks overlap more than one Hi-C fragment and vice versa), 12,380 (85%) have over 50% of peak bases within the Hi-C fragment and 10,329 (71%) have the entire peak within the Hi-C fragment (S12 Table), suggesting that we would obtain similar results using more strict overlap thresholds. Of the 14,080 peaks, 3,436 were preadipocyte-dependent and 4,873 were adipocyte-dependent (5,771 were context-dependent but not preadipocyte- or adipocyte-dependent, hereafter called 'other context-dependent peaks'). We identified more links for adipocyte peaks than for preadipocyte peaks, consistent with our use of Hi-C data only from mature adipocytes, not preadipocytes. Most distances from peaks to gene TSS linked by Hi-C (85%) were within 500 kb, and 97% were within 1.2 Mb (Fig 2D).

In the second approach, we identified context-dependent peaks that overlapped adipose eQTL signals [5], defining each signal as all variants in high linkage disequilibrium with a lead eQTL variant ($r^2$>0.8, Fig 2B and S12 Table). Of 3,002 peaks linked to 2,369 genes (3,794 peak-gene pairs), 805 linked from preadipocyte-dependent peaks and 996 from adipocyte-dependent peaks (1,201 linked from other context-dependent peaks). The larger number of links from adipocyte peaks than preadipocyte peaks is consistent with use of eQTL from adipose tissue, which contains more adipocytes than preadipocytes [16,31]. We identified 4,549 adipose eQTL variants within the context-dependent peaks; these variants could be part of the mechanisms regulating expression level of the corresponding genes. Most distances from peaks to gene TSS linked by eQTL (87%) were within 200 kb, and all were within 1 Mb, the distance threshold used for the eQTL study (Fig 2D).

In the third approach, we identified context-dependent peaks linked to a gene by Hi-C or eQTL overlap for which the gene also showed context-dependent expression between any timepoint comparison (Fig 2C and S12 Table). Of the 14,080 peaks identified by Hi-C, 4,462 peaks also linked to a context-dependent gene (1,000 linked from preadipocyte-dependent peaks and 1,681 linked from adipocyte-dependent peaks, 1,781 linked from other context-dependent peaks). Of the 3,002 peaks identified by eQTL, 720 contained a context-dependent gene (134 linked from preadipocyte-dependent peaks, 298 linked from adipocyte-dependent peaks, 288 linked from context-dependent but not preadipocyte- or adipocyte-dependent peaks) (S12 Table).

Each approach to link regulatory peaks to genes can add an additional level of evidence to support the predicted gene target. We next identified peaks linked to the same gene through more than one approach. Of 16,076 total peaks linked to a gene through at least one of the three approaches (S12 Table), 78 peaks were linked to the same gene through all three approaches and 5,145 peaks were linked to the same gene through two or more approaches (Fig 2E). Of the 78 peaks linked to 59 genes through all three approaches, interesting candidate regulatory elements include four peaks linked to *CDKN2B*, whose gene product has known roles in cell cycle control and whose regulation has been linked to coronary artery disease [36,37]. Of the 5,145 peaks linked to 1,670 genes through at least two approaches, 1,143 linked from preadipocyte-dependent peaks and 1,945 linked from adipocyte-dependent peaks (2,057 linked from other context-dependent peaks). Although peaks linked by all three approaches have the most supporting evidence, to prevent overlooking interesting candidates we considered peaks linked by two or more methods when evaluating candidates for functional evaluation.

## Trait heritability enrichment within context-dependent peaks

We used stratified LD score regression [38] to compare heritability enrichment for selected cardiometabolic traits in preadipocyte-dependent peaks, adipocyte-dependent peaks, and bulk adipose tissue peaks. Given that preadipocyte-dependent and adipocyte-dependent peaks cover a small portion of the genome (~0.45%), we also ran stratified LD score regression on the top 100,000 consensus peaks (ranked by median peak p-value across replicates) in each SGBS differentiation day. For comparison, we also ran stratified LD score regression using the adipose tissue peaks.

Different traits were enriched in adipocyte-dependent and preadipocyte-dependent peaks. For waist-hip ratio adjusted for BMI (WHRadjBMI), we observed significant enrichment for adipocyte-dependent peaks (z-score = 4.7, $P<1.2 \times 10^{-6}$) and adipose tissue peaks (z-score = 5.2, $P<1.0 \times 10^{-7}$) but not for preadipocyte-dependent peaks (z-score = -1.1, $P<0.86$) (Fig 3A and S13 Table). Results were consistent for the top 100,000 consensus peaks in each SGBS differentiation day; the modest enrichment in D0 peaks could be partly due to peaks shared between timepoints. We also observed nominal enrichment for HDL heritability in adipocyte-dependent and adipose tissue peaks. In contrast, we observed significant enrichment for coronary artery disease (CAD) heritability in SGBS D0 (z-score = 2.8, $P<2.9 \times 10^{-3}$) and adipose tissue (z-score = 3.3, $P<5.4 \times 10^{-4}$), with weaker and still nominally significant enrichment in D4 (z-score = 2.4, $P<9.0 \times 10^{-3}$) and D14 (z-score = 2.3, $P<0.01$); the lack of enrichment in preadipocyte-dependent and adipocyte-dependent peaks may be due to their low genomic coverage. All peak sets showed less heritability enrichment relative to baseline for rheumatoid arthritis, a negative control, except for adipocyte-dependent peaks, which showed nominal enrichment (z-score = 1.8, $P<0.04$), suggesting that adipocytes may have moderate relevance for this trait. We did not observe enrichment of BMI heritability in any peak set, consistent with our previous finding that BMI GWAS loci were not enriched in adipose tissue or SGBS peaks [14] and with findings from other studies that BMI loci are enriched in central nervous system cell types and pathways [38,39]. A complementary approach using all traits in the GWAS catalog [40] grouped by Experimental Factor Ontology terms showed similar results (Fig 3B and S14 Table). The most enriched terms for adipocytes included waist-hip ratio, cholesterol, inflammatory traits, and birthweight, whereas the most enriched terms for preadipocytes included atrial fibrillation and inflammatory traits. We also observed enrichments for traits with less apparent, but established connections to cardiometabolic traits, including forced expiratory volume, a measure of lung function that has been shown to be lower in individuals with metabolic syndrome and high body fat percentage [41,42], and intraocular pressure, which has been shown to be higher in individuals with metabolic syndrome and markers of obesity [43,44]. Taken together, we found that peaks in adipocytes contribute more to heritability of WHRadjBMI, whereas preadipocytes may contribute more to heritability of CAD, though to a lesser degree.

## Fine-mapping of GWAS variants using context-dependent peaks and allelic imbalance

To identify genetic variants that may have context-dependent effects on disease-relevant traits, we identified distinct signals from the GWAS catalog [40] (see Methods) for which a proxy variant (LD $r^2>0.8$) is located within a context-dependent peak. Of 4,954 context-dependent peaks that overlapped GWAS signals, 1,448 were preadipocyte-dependent and 1,461 were adipocyte-dependent (S15 Table).

At some GWAS loci, these context-dependent peaks can be linked to genes. We observed 4,284 peak-gene pairs that overlapped GWAS variants, and 799 of these pairs, representing

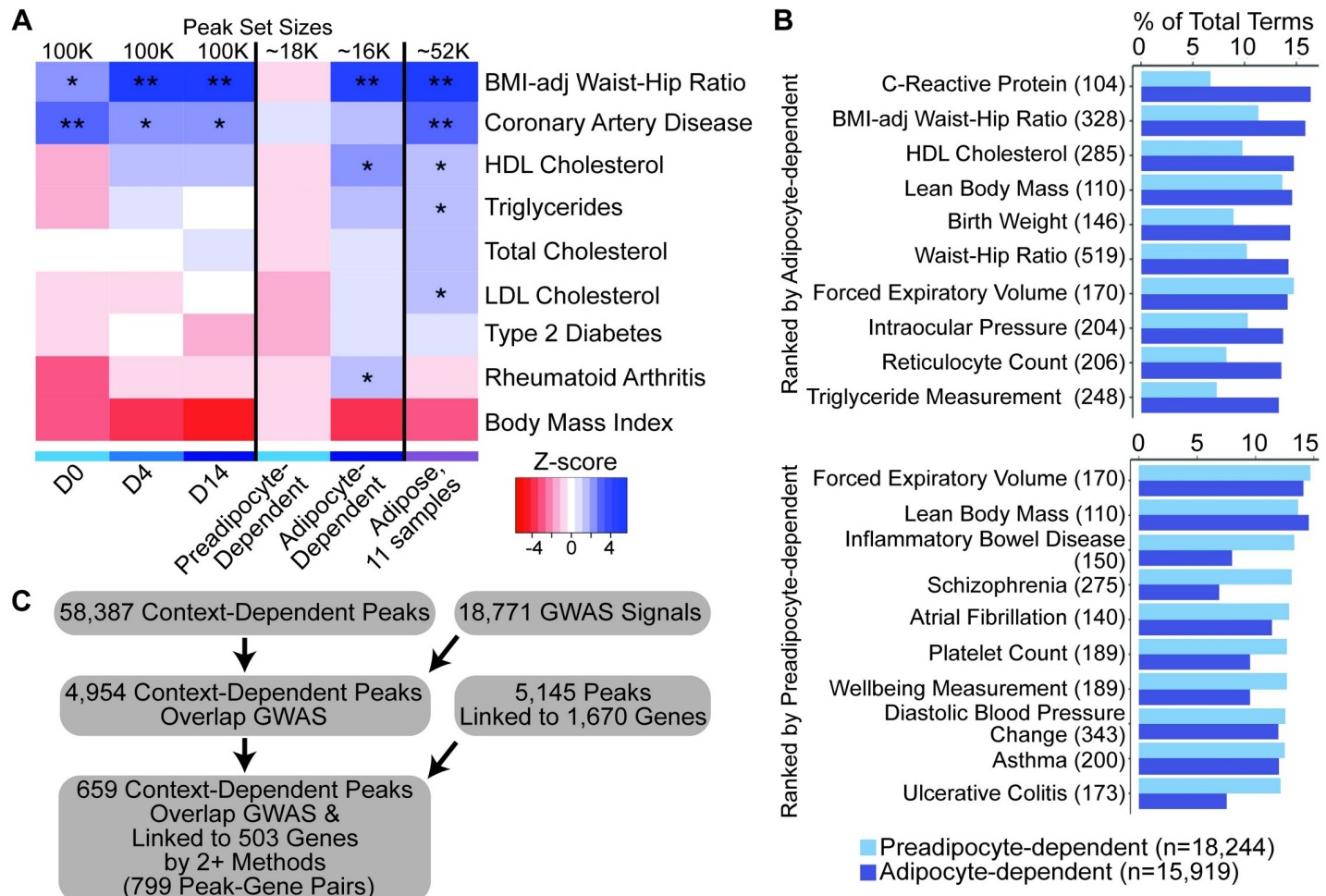

**Fig 3. Linking peaks to GWAS signals.** (A) Heatmap of cardiometabolic trait GWAS locus enrichment; rheumatoid arthritis was selected for comparison. Peak sets include 100,000 peaks from individual days, preadipocyte- and adipocyte-dependent peaks derived from pairs of timepoints, and adipose tissue peaks. Values in S13 Table. **, P < 0.0056; *, P < 0.05 (B) Barplots of normalized counts of specific experimental factor ontology (EFO) terms for GWAS signals with a variant in a context-dependent peak. Barplots show the top ten EFO terms ranked by normalized count for either preadipocyte-dependent peaks, or adipocyte-dependent peaks. Total number of signals for each term used in the overlap is noted in parentheses in the axis label. Total number of signals for each term overlapping a context-dependent peak is noted to the right of the "All Context-dependent" bar. Values in S14 Table. (C) Flowchart identifying context-dependent peaks overlapping GWAS signals and linked to genes through 2 or more methods.

659 unique peaks, were supported by two or more approaches (Fig 3C and S12 Table). Of these 659 peaks, 265 were adipocyte-dependent, 143 were preadipocyte-dependent, and 251 were other context-dependent peaks. Of these 659 peaks, 191 (29%) overlapped adipose tissue peaks, which generally had weaker signals than the SGBS peaks. At one locus, we identified two peaks more accessible in adipocytes that overlap adipose eQTL variants for *ADIPOQ* (peak96641: rs76071583; peak96640: rs143257534), which also showed adipocyte-dependent expression. These peaks also overlap adipose consensus peaks and GWAS variants associated with adiponectin levels [45], including rs76071583, previously shown to exhibit allelic differences in binding of the transcription factor CEPB-α and transcriptional activity in adipocytes [46]. *CEBPA* has higher expression in adipocytes than preadipocytes (D4>D0 LFC = 8.6, D14>D0 LFC = 9.2, S10 Table), consistent with the context-dependent regulatory effect.

To identify GWAS variants that may alter chromatin accessibility at different stages of differentiation, we also identified variants exhibiting allelic imbalance (AI) in ATAC-seq reads

across SGBS technical replicates. Because SGBS cells originate from one individual, we could only test for AI at heterozygous variants in one individual. We identified 574, 996, and 489 variants showing significant AI (FDR<5%) on D0, D4, and D14, respectively, and 582 AI variants in 454 context-dependent peaks (S16 Table), including 90 peaks that harbored more than one AI variant. Of the 454 context-dependent peaks, 64 were linked to a target gene by two approaches, 55 contained GWAS variants that exhibited AI, and 13 linked to both a target gene and GWAS variant. At an example with both types of data, a variant (rs11039149) that showed significant AI in days 4 and 14 was found within a peak more accessible in D4 compared to D0 (peak23801) and is an eQTL variant for the adipocyte-dependent gene *NR1H3* (S12 and S16 Tables and S6 Fig). The more accessible allele rs11039149-G is associated with lower *NR1H3* expression. rs11039149 is a GWAS variant for HDL cholesterol [47] and proinsulin [48]. NR1H3 has previously been shown to be involved in lipid transport [49], and one or more of these variants could alter *NR1H3* expression and affect associated metabolic traits. Combining ATAC AI, context-dependent peaks, and target genes helps connect variants to regulatory elements and genes and can identify variants with context-dependent effects on gene regulation.

## Functional evaluation of candidate regulatory elements reveals context- and allele-dependent mechanisms

Of the 659 context-dependent peaks that we linked to target genes and GWAS signals, we tested two for allele-dependent effects on transcriptional activity using reporter gene assays in SGBS preadipocytes and 12-day differentiated adipocytes. At a first GWAS locus for palmitoleic acid [50], we identified an adipocyte-dependent peak (Fig 4A, peak19405; D4>D0: LFC = 3.8; D14>D0: LFC = 2.8) that we linked to the gene *SCD*, encoding Stearoyl-CoA Desaturase, through two approaches, overlap of the peak with an adipose eQTL variant (rs603424, P = 1.6x10$^{-9}$) associated with *SCD* [5] and adipocyte-dependent expression of SCD (D4>D0: LFC = 6.3; D14>D0: LFC = 8.2) (Fig 4A). *SCD* codes for an enzyme involved in fatty acid synthesis [51]. Peak19405 also overlaps a consensus adipose tissue peak and contains rs603424, the G allele of which is associated with higher *SCD* expression [5] and higher palmitoleic acid [50]. We tested a 592-bp region spanning the majority of peak19405 for allele-dependent functional effects. In adipocytes, the construct containing the rs603424-G allele demonstrated significantly increased transcriptional activity compared to the construct containing the rs603424-A allele (forward P = 0.003, reverse P = 0.0001; Figs 4B and S7 and S17 and S18 Tables), consistent with the direction of effect observed in the adipose eQTL. Together, these results suggest that in adipocytes but not preadipocytes, rs603424-G increases transcriptional activity of *SCD* to increase palmitoleic acid levels.

At a second GWAS locus for type 2 diabetes [52], we identified two candidate regulatory elements and tested both for allele-dependent effects on transcriptional activity. One candidate is an adipocyte-dependent peak (Fig 5A, peak81750, containing rs55966194, D4>D0: LFC = 4.2 and D14>D0: LFC = 3.1) that we linked to *EYA2*, encoding Eyes Absent Transcriptional Coactivator and Phosphatase 2, through colocalization with an adipose eQTL (rs55966194, P = 6.0x10$^{-10}$) [5] and adipocyte-dependent expression of the linked gene (D4>D0: LFC = 1.7; D14>D0: LFC = 1.4) (Fig 5A). *EYA2* codes for a protein that has been linked to adipocyte lipolysis [53]. Also, at this locus, a second candidate regulatory element is an adipose-specific peak not detected in SGBS and which contains variant rs59791349, which is a proxy variant for an adipose eQTL for *EYA2* [5] and a GWAS locus for type 2 diabetes [52]. The C alleles for both rs55966194 and rs59791349 are associated with higher *EYA2* expression and increased risk of type 2 diabetes. We tested regions spanning the majority of

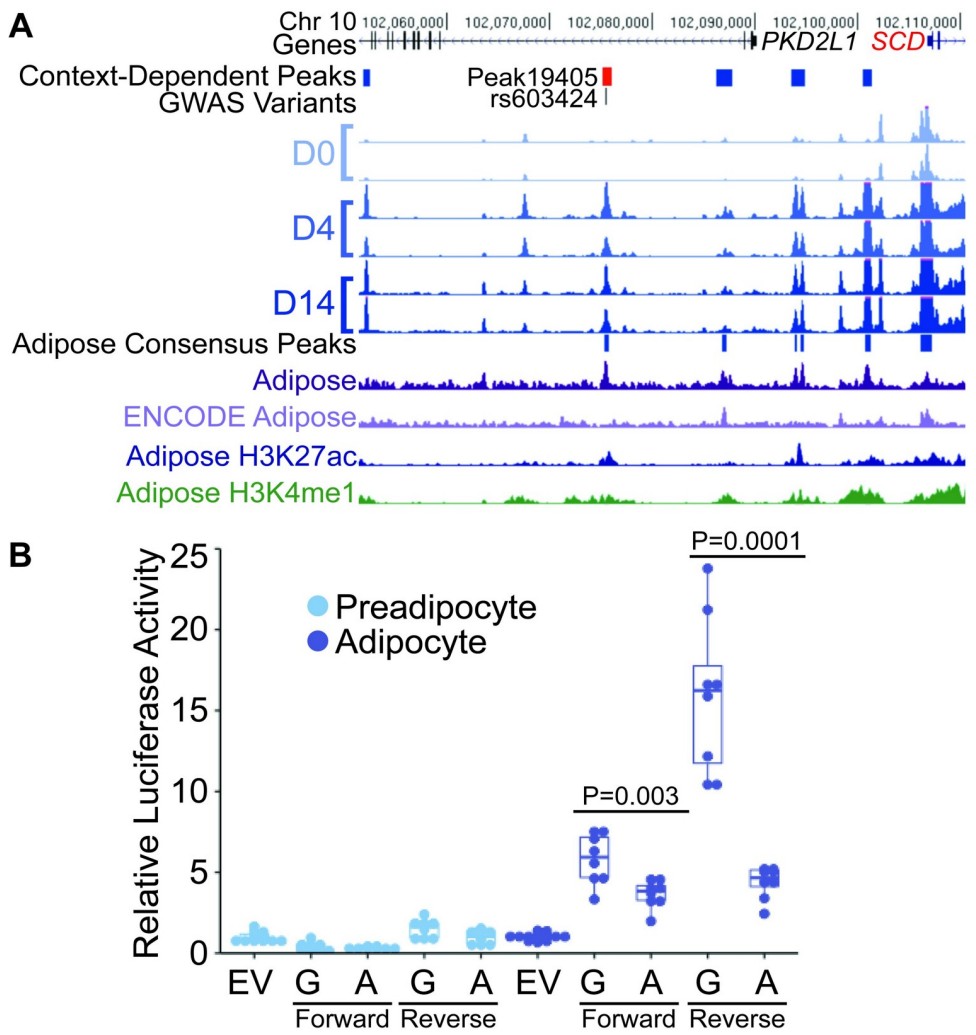

**Fig 4. Allelic differences in transcriptional activity for a context-dependent regulatory variant in a context-dependent element at the *SCD* locus.** (A) Peak19405 (red) is more accessible in D4 and D14 adipocytes than D0 preadipocytes, overlaps an adipose tissue consensus peak (dark purple), and overlaps variant rs603424, which is associated with blood plasma levels of palmitoleic acid and adipose *SCD* expression. *SCD* is also more highly expressed at D4 and D14 compared to D0. Additional tracks show adipose tissue ATAC-seq from ENCODE (light purple) and adipose nuclei histone mark ChIP-seq from the Roadmap Epigenomics project (blue and green). (B) A 592-bp genomic region surrounding peak19405 containing the rs603424-G allele shows increased transcriptional activity compared to the rs603424-A allele in the forward and reverse orientations only in adipocytes (tested at day 12), the context in which chromatin was more accessible compared to preadipocytes. Dots represent two independent constructs assayed from four replicates each. Luciferase activity was normalized relative to an empty vector (EV). Values in S18 Table.

each peak for allele-dependent transcriptional activity. The 419-bp region for adipocyte-dependent peak81750 containing the rs55966194-C allele demonstrated modest allelic differences only in the reverse orientation (P = 0.06, Figs 5B and S8 and S17 and S18 Tables), whereas the 288-bp region for the adipose peak containing rs59791349-C demonstrated significantly higher transcriptional activity than the rs59791349-T allele in both orientations and both cell types (adipocytes forward P = 0.0029, adipocytes reverse P = 0.0058; preadipocytes forward P = 0.0008, preadipocytes reverse P = 0.0015; Figs 5C and S8 and S17 and S18 Tables). The allelic differences in transcriptional activity were consistent with the direction of effect of

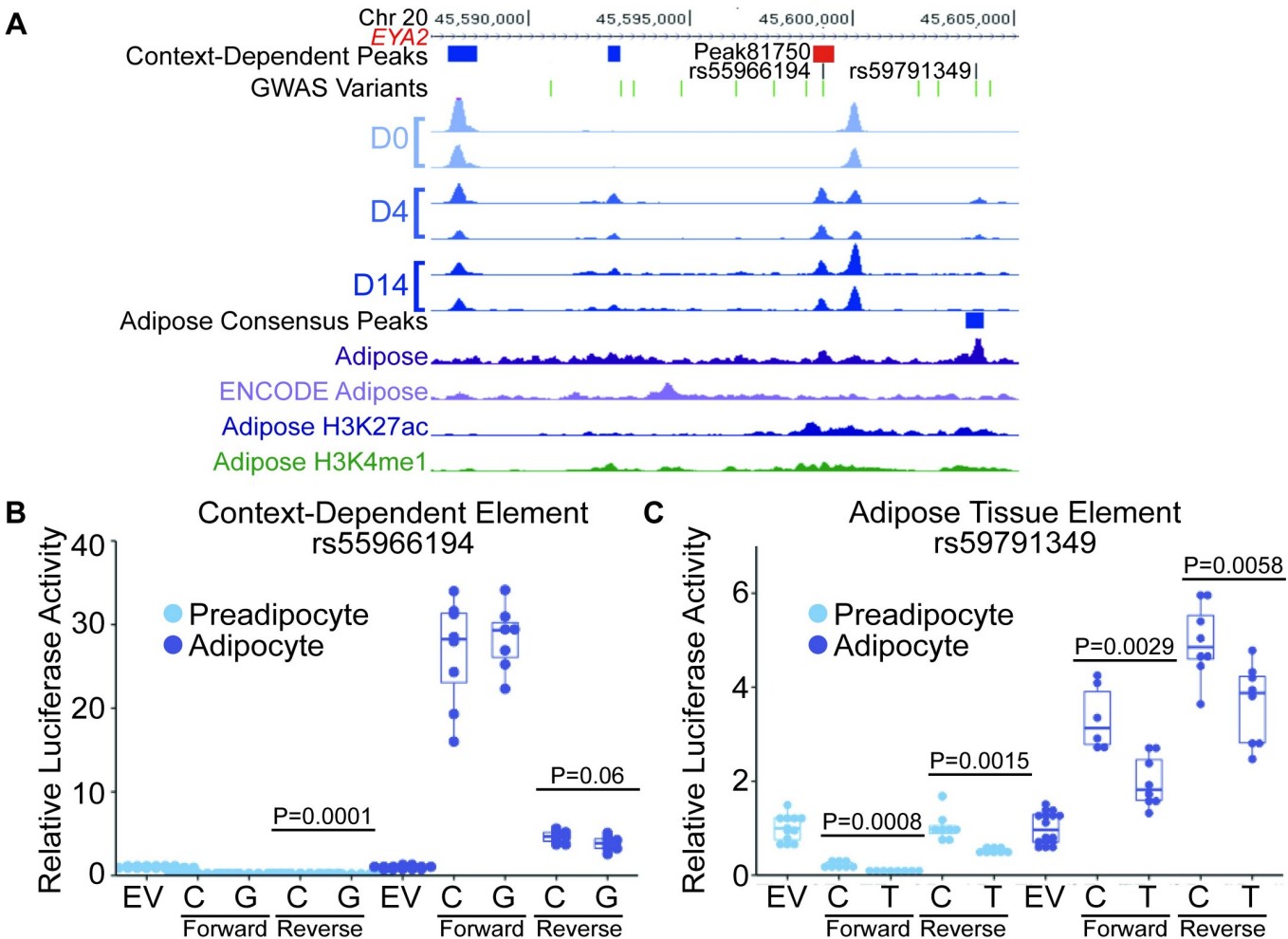

**Fig 5. Allelic differences in transcriptional activity for variants in two regulatory elements at the *EYA2* locus.** (A) Peak81750 (red) is more accessible in D4 and D14 adipocytes and overlaps variant rs559066194, which is associated with increased risk of type 2 diabetes and increased *EYA2* expression. EYA2 is more highly expressed at D4 and D14, compared to D0. A second variant at this locus, rs59791349, intersects a consensus adipose peak (dark purple) but not a context-dependent peak. Additional tracks as in Fig 4. (B-C) Values in S18 Table. (B) A 419-bp genomic region surrounding peak81750 containing the rs555966194-C allele shows modestly-increased transcriptional activity compared to the rs555966194-G allele in the reverse orientation, but not the forward, in adipocytes (tested at day 9), the context in which chromatin was more accessible compared to preadipocytes. Dots represent two independent constructs assayed from four replicates each. Luciferase activity was normalized relative to an empty vector (EV). (C) A 288-bp genomic region containing the rs59791349-C allele shows increased transcriptional activity compared to the rs59791349-T allele in both orientations and in both preadipocytes and adipocytes (tested at day 9). Dots represent two independent constructs assayed from four replicates each. Luciferase activity was normalized relative to an EV.

the adipose eQTL. These results suggest that in both preadipocytes and adipocytes, rs59791349-C increases transcriptional activity of EYA2 to increase risk of diabetes. Altogether, the experiments at these two loci demonstrate that context-dependent peaks can, but do not always, predict allele-dependent transcriptional activity, as other mechanisms may be involved. These results also suggest the value of using both cell type-specific and tissue-derived regulatory elements to identify functional regulatory variants.

## Discussion

In this study, we generated chromatin accessibility and gene expression profiles for preadipocytes, immature adipocytes, and mature adipocytes and identified context-dependent peaks

during adipocyte differentiation as candidate regulatory elements. We linked these regulatory elements to candidate genes using three approaches and identified context-dependent regulatory elements at GWAS loci. Our consensus subcutaneous adipose tissue peak map based on profiles from 11 individuals provided a resource to expand on existing human adipose peak maps [19,54,55] and to prioritize among peaks from the SGBS cell model. Finally, we identified 659 context-dependent regulatory elements at GWAS loci that were linked to genes and showed through functional tests that elements can exhibit context-dependent allelic differences in transcriptional activity, identifying plausible disease mechanisms.

Chromatin accessibility profiles differ between samples for biological and technical reasons. Biological reasons can include cell type and cell context. A technical source of variation between our profiles could be due to heterogenous sequencing protocols with a mix of paired-end, single-end, and variable read lengths. We addressed the heterogenous sequencing protocols in our analyses as described in Methods, but it could contribute to differences between libraries. Reassuringly, our SGBS ATAC libraries cluster by day despite differences in sequencing parameters. Additionally, while SGBS cells are a useful human adipocyte model, some aspects of the chromatin accessibility profile could be due to the cells growing in culture or the overgrowth syndrome disease state that allows the cells to grow without being transformed. To address these limitations, we identified SGBS peaks that overlapped adipose tissue peaks, including the peak we tested at the *SCD* locus. Although it remains challenging to compare between species, we observed enrichment of motifs for well-known adipogenesis transcription factors CEBP, PPAR, and RXR within adipocyte-dependent peaks, consistent with a study of changes during adipogenesis in a 3T3-L1 mouse cell line [20].

Our differential analyses of peaks and gene expression profiles between timepoints suggest that most peak changes occur between D0 and D4 and remain stable between D4 and D14, while a larger proportion of gene expression changes occur between D4 and D14. The observation that peak changes occur early and remain largely stable is consistent with a previous study that found a majority of chromatin accessibility changes in a 3T3-L1 mouse-derived adipocyte cell line occurred between two and four hours after the initiation of differentiation [20]. The observation that gene expression may continue to change throughout later stages of differentiation is consistent with a study that showed gene expression changing between 7-day intervals up to day 21 in human adipose-derived stromal cells [32]. After initial analyses suggested that few context-dependent peaks arose between D4 and D14, we investigated chromatin accessibility at an earlier timepoint of immature adipocytes differentiated for two days (D2). Preliminary analysis of D2 also showed that no peaks were differential between D2 and D4, so we did not generate further D2 data. Similarities between D4 and D14 also led us to focus on the subsets of context-dependent peaks that were specific to preadipocytes (D0>D4 and D0>D14) or adipocytes (D4>D0 and D14>D0), rather than the limited number that were specific to mature adipocytes (D14>D0 and D14>D4).

We used two approaches to link context-dependent peaks to genes: overlap with existing adipocyte promoter capture Hi-C regions and with known adipose eQTL variants, and we determined which of these linked genes also showed expression differences between differentiation timepoints. Promoter capture Hi-C has the advantage of identifying direct connections between regulatory elements and genes, even over large distances. However, physical proximity does not necessarily imply a regulatory relationship, and the data we used was for adipocytes, not preadipocytes, and therefore could have detected connections for the D4 and D14 timepoints better than for D0. While most promoter capture Hi-C fragments have high resolution (median ~3 kb in the analyzed dataset), the location of restriction sites in the genome limits resolution for some fragments (~10% of fragments had >10 kb resolution). Our second approach based on overlap with adipose eQTL variants has the advantage that the identified

variants are associated with differences in gene expression. Two disadvantages of the eQTL approach are that eQTL studies may be underpowered, so not all associations are discovered, and that the adipose tissue used in the eQTL study is comprised of multiple cell types, not only adipocytes. Although adipose tissue is heterogenous, it is known to contain more adipocytes than preadipocytes [16,31]. Therefore, the eQTL method also could have detected connections for the D4 and D14 timepoints better than for D0. Peaks linked to genes by eQTL tended to be closer to the TSS of the linked gene compared to Hi-C, partially due to the shorter distance window used in the eQTL data than the Hi-C data. To incorporate differential gene expression into the identification of peak-to-gene links, we initially considered using proximity between context-dependent peaks and context-dependent genes. However, proximity is indirect and requires selecting an arbitrary threshold for maximum distance between peak and gene. Thus, we used context-dependent gene expression as additional supporting evidence for links made by other methods. Although indirect, context-dependent genes have the advantage of being observed in the same cell model and at the same timepoints, and can help determine if a regulatory element has a positive or negative effect on gene expression. Due to the advantages and disadvantages of the different approaches, the largely different peak-to-gene links detected were not surprising. Using multiple approaches to link regulatory elements to candidate genes can overcome the limitations of each approach, and genes identified by multiple methods can increase confidence, although genes linked by even a single method merit further investigation.

We used AI in SGBS ATAC-seq reads to provide suggestive evidence that GWAS variants may alter chromatin accessibility at different stages of adipocyte differentiation. Although we only tested AI at heterozygous variants from one individual, which limits heterozygous sites available for testing, we identified 55 peaks containing GWAS variants that exhibited AI, 13 of which were linked to genes. ATAC-seq in additional cell lines with diverse genotypes would improve the ability to detect AI. Previous studies have mapped AI and chromatin accessibility QTL in different contexts [9,56,57], which allowed for testing of more variants and identification of more robust context-dependent genetic effects on gene regulation. Our results demonstrate that AI in ATAC-seq reads from one individual can be used to predict regulatory variants, although identifying AI in larger sample sizes would lead to more comprehensive and robust results and more genetic variants.

We followed up context-dependent regulatory elements at two GWAS loci by testing variants for effects on context-dependent transcriptional reporter gene activity. Due to the bias towards adipocytes of our methods to link peaks to genes, we focused on regulatory elements more accessible in adipocytes. At *SCD*, we observed consistent evidence of context- and allele-dependent transcriptional activity among technical replicates. The regulatory element that was more accessible in adipocytes contained an allele associated with increased adipose tissue expression of *SCD* [5] and increased palmitoleic acid [50]. *SCD* codes for an enzyme involved in fatty acid synthesis [51], therefore increased *SCD* expression is a likely mechanism to increase palmitoleic acid levels. In reporter assays, the element showed higher transcriptional activity in adipocytes than preadipocytes, and the allele associated with higher adipose *SCD* expression showed higher transcriptional activity, only in adipocytes. These data suggest that the regulatory element we identified increases *SCD* expression to increase palmitoleic acid levels in adipocytes.

At the second locus we examined, *EYA2*, the results are more complex. We identified two candidate regulatory elements, one that was adipocyte-dependent and one that was present in the consensus adipose tissue map. Both regulatory elements contained variants associated with adipose tissue expression of *EYA2* [5] and type 2 diabetes [52]. Both regulatory elements demonstrated higher expression of the reporter gene in adipocytes than preadipocytes, consistent

with the context in which one element was more accessible and with the large proportion of adipocytes in adipose tissue [16,31]. However, only the consensus adipose element demonstrated clear allele-dependent transcriptional activity. This result demonstrates that, while identifying loci with context-dependent peaks linked to genes and traits still is useful for identifying candidates, it does not mean the identified variant is responsible. However, the variant within the adipocyte-dependent peak at this locus may still exhibit allelic effects on regulatory activity that are not detectible in *in vitro* transcriptional reporter assays. For the *EYA2* locus, our adipose consensus map guided us to investigate an additional candidate regulatory element that demonstrated an allele-dependent effect on transcriptional activity. *EYA2* codes for a transcriptional coactivator that has been linked to many developmental processes and adipocyte lipolysis, consistent with a role in adipocyte biology and metabolic traits [53,58]. Our reporter assays demonstrate allelic differences in transcriptional activity for elements at two loci, however, additional experiments are needed to validate specific regulatory elements within these peaks in the context of chromatin accessibility and the effect on regulation on the predicted gene.

This study extends our previous study that reported ATAC-seq peaks in SGBS cells and adipose tissue from three individuals [14]. Genetic variation contributes to differences in peaks, so we profiled adipose tissue in additional individuals to capture peaks that could have been missed in fewer samples due to genetic variants or environmental/physiological differences between individuals. In general, ATAC-seq data from frozen adipose tissue demonstrated lower quality than our SGBS preadipocytes and other frozen tissues [24,59,60], despite our efforts to optimize library preparation with different buffers, detergents, and ratios of transposase to nuclei. Freezing has been shown to affect ATAC-seq library quality and comparisons of ATAC-seq profiles in samples using various freezing methods suggest cryopreserved tissue demonstrated higher quality than flash-frozen tissue [59]. High lipid content could also have affected adipose tissue profile quality, as adipose tissue has a high ratio of adipocyte cells [16,31] and lipid content somewhat affected ATAC-seq in cultured SGBS cells, as fewer D14 adipocyte samples met QC thresholds compared to D0 and D4 cells, despite being cultured and processed in parallel. The consensus map of adipose peaks based on the 11 samples of at least moderate quality showed similar overlap with adipocyte nuclei promoters and enhancers as our previous map based on three samples, but the inclusion of additional samples should make the 11-sample consensus map more robust.

Overall, we demonstrated that context-dependent chromatin accessibility identifies context-dependent regulatory elements that can aid understanding of mechanisms behind cardiometabolic traits. By identifying adipocyte differentiation context-dependent regulatory elements and linking them to genes and GWAS traits, we filtered from 58,387 context-dependent regulatory elements to 659 elements with a candidate mechanism. Additional study of these regulatory elements could lead to a better understanding of the role of adipocytes and adipocyte differentiation in cardiometabolic disease traits as well as other relevant traits we identified through enrichment analyses such as lung function. This could also be applied to other adipocyte contexts to identify additional context-dependent mechanisms.

## Methods

### Ethics statement

The Ethics Committee of the University of Eastern Finland in Kuopio and the Kuopio University Hospital approved the METSIM study and it was carried out in accordance with the Helsinki Declaration. Formal written consent was obtained from METSIM participants.

## Cell culture

SGBS cells [21] were generously provided by Dr. Martin Wabitsch (University of Ulm) and cultured as previously described [61]. Briefly, we cultured SGBS preadipocytes in serum-containing basal medium (DMEM:F12 + 33uM biotin + 17uM pantothenate) with 10% FBS until confluent, then rinsed in phosphate-buffered-saline (PBS) and differentiated for four days in medium supplemented with 0.01 mg/mL transferrin, 20 nM insulin, 200 nM cortisol, 0.4 nM triiodothyronine, 50 nM dexamethasone, 500 uM IBMX, and 2 uM rosiglitazone. After four days, we maintained differentiated SGBS cells in basal medium supplemented with 0.01 mg/mL transferrin, 20 nM insulin, 200 nM cortisol, 0.4 nM triiodothyronine until harvested. HEK293T cells (ATCC, Manassas, VA) were grown in DMEM supplemented with 10% FBS.

## Adipose tissue

Human subcutaneous abdominal adipose tissue biopsies were obtained from METabolic Syndrome in Men (METSIM) [23] participants as previously described [4]. Adipose tissue was obtained through either a needle or surgical biopsy and flash frozen and stored at -80°C until use.

## ATAC-seq library preparation

We profiled chromatin accessibility in SGBS cells at D0, D4, and D14 of adipocyte differentiation following the omni-ATAC protocol [24] using unique, dual-barcoded indices. We isolated nuclei and used a cell countess to aliquot 50,000 nuclei per library. After initial optimization of Tn5:nuclei ratios, we proceeded with 5 uL of Tn5 per library, some early libraries were prepared with 2.5 uL of Tn5 as indicated (S1 Table). For adipose tissue samples we used the original or omni-ATAC protocol [11,24] as indicated (S7 Table). We cleaned the transposase reaction and final library with Zymo DNA Clean and Concentrator (D4029). We visualized and quantified libraries using a TapeStation, and sequenced with paired-end or single-end reads on a Highseq or Novaseq as indicated (S1 and S7 Tables).

## ATAC-seq read alignment and peak calling

For METSIM samples, ATAC-seq read lengths ranged from 50–150 bp, depending on sequencing center, so all libraries were trimmed to a uniform length of 50 bp before processing. Three METSIM ATAC-seq libraries were single-end and were processed with a single-end version of the following pipeline. All other libraries were paired-end. SGBS ATAC-seq reads were not length-trimmed before processing, although some libraries had 50bp reads and others had 150bp reads. We trimmed sequencing adapters and low quality base calls from the 3' ends of reads using cutadapt [62] with parameters -q 20 –minimum-length 36. We aligned trimmed reads to the hg19 human genome [63] using bowtie2 [64] with parameters–minins 36 –maxins 1000 –no-mixed–no-discordant–no-unal and selected nuclear chromosomal alignments with mapq>20 using samtools [64]. We removed alignments overlapping high-signal regions (Duke excluded and ENCODE/DAC exclusion list regions) [65] using BEDTools pairToBed [66] with the parameter -type notospan. We removed duplicate alignments using Picard MarkDuplicates (https://github.com/broadinstitute/picard) and generated ATAC-seq quality metrics using ataqv [67]. Ataqv is only designed for paired-end reads, so we used a customized approach to calculate TSS enrichment for the single-end METSIM libraries. To calculate TSS enrichment, we generated 2,001-bp windows containing the TSS and 1 kb flanking regions on either end for the set of 5,307 RefSeq housekeeping TSSs used by ataqv for TSS enrichment. We then calculated the number of ATAC reads overlapping each base within

these 2,001 bp windows for each METSIM sample using BEDTools coverage with the -d option and made a matrix of coverage for these windows using python. Finally, we summed the coverage across each TSS window within the same sample and calculated TSS enrichment by dividing the summed coverage at the TSS by the mean summed coverage of the 100 bases at the leftmost and rightmost ends of the windows using R.

Prior to peak calling, we trimmed alignments so their 5' ends corresponded to the Tn5 binding site (+4 for + strand alignments and -5 for–strand alignments) [11] and smoothed signal by extending alignments 100 bp on either side of the Tn5 binding sites using BEDTools slop [52]. We called peaks (FDR<5%) with MACS2 [68] with parameters -q 0.05 –nomodel–bdg and generated ATAC signal bigwig files from MACS2 bedGraph files using the bedGraphToBigWig tool from ucsctools [69]. For SGBS libraries, we proceeded with analyses on a final set of libraries that met our signal-to-noise quality thresholds with a fraction of reads in peaks (FRiP) greater than 20% and a transcription start site enrichment greater than 5 [18]. For METSIM libraries, we selected libraries that had TSS enrichment > = 4 calculated from our customized script that works on single-end and paired-end reads. TSS enrichment values produced by our script are generally higher than those calculated by ataqv, and TSS enrichment of 4 from our script corresponds roughly to TSS enrichment of 3 from ataqv.

For each analyzed day of SGBS differentiation, we generated a set of consensus ATAC peaks using the following method. First, we merged peak genomic coordinates across replicates for a given day using BEDTools merge [66]. Second, we defined consensus peaks as merged peaks that overlapped individual replicate peaks in greater than 50% of replicates (at least 3 out of 5 replicates for D14 and 6 out of 10 replicates for D0 and D4).

## Identification of differentially accessible peaks

We generated a set of merged peaks to test for differential chromatin accessibility by merging the top 100,000 consensus peaks in each day (ranked by median peak p-value across replicates). We quantified the accessibility of these merged peaks in each library using feature-Counts [70]. We computed the GC percent of each peak using BEDTools nuc [66] and generated within-library GC bias normalization factors using full quantile normalization with EDASeq [71]. We then used EDASeq GC bias normalization factors within DESeq2 [72] and used DESeq2 size factors to control for differences in sequencing depth between libraries. We tested for differential chromatin accessibility using DESeq2 [72] and classified peaks with FDR<5% and log fold change (LFC)>1 as significantly differential.

## Enrichment of transcription factor motifs in differential peaks

We tested for enrichment of 319 transcription factor (TF) motifs in adipocyte or preadipocyte-dependent peaks using the findMotifsGenome tool from HOMER [73] with the -size 200 option. We used peaks that were not differential in any pairwise day comparison (FDR>50%, absolute value of LFC<1) as background in the enrichment analyses. We classified motifs with a p-value less than the Bonferroni-corrected threshold of $1.6 \times 10^{-4}$ (0.05/319 motifs) as significant.

## Gene ontology enrichment of genes near differential peaks

We tested if genes near adipocyte and preadipocyte-dependent peaks were enriched for specific biological processes using the Genomic Regions Enrichment of Annotations Tool (GREAT) web tool (http://great.stanford.edu/public/html/) [74] with the GO Biological Process ontology [74,75]. We ran GREAT version 4.0.4 with the default parameters of basal plus extension, proximal 5 kb upstream to 1 kb downstream, distal 1000 kb (1 Mb), and a whole

genome background. We classified ontology terms with Minimum Region-based Fold Enrichment> = 2 and FDR<5% as significantly enriched.

## Identification of adipose tissue consensus peaks

We constructed an initial set of adipose tissue consensus peaks using the 17 METSIM libraries with TSS enrichment> = 4 (our customized TSS enrichment script). To construct consensus peaks, we took the union of peaks across all 17 samples and selected union peaks that overlapped (shared at least one base) with a peak in 3 samples. To identify outlier samples, we computed PCA of ATAC-seq read counts within consensus peaks and performed hierarchical clustering of the top 10 PCs (S3 Fig). We identified 6 outlier samples: four samples were generated with the omni-ATAC protocol [24] (whereas all other samples were generated using the original ATAC protocol [11]), one sample had a much higher percentage of mitochondrial reads compared to other samples, and one sample had substantially fewer peaks compared to other samples. Adipose tissue peaks from the 11-sample peak set showed stronger overlap with Roadmap Epigenomics adipocyte nuclei enhancers (Figs 1D and S3D) and stronger enrichment for all tested traits except BMI (Figs 3 and S3E) compared to the 17-sample set. Therefore, we removed these 6 samples and generated consensus peaks with 11 samples, using the same approach as for 17 samples.

## RNA-seq library preparation, read alignment, and identification of differentially expressed genes

We isolated total RNA from SGBS cells at D0, D4, and D14 of differentiation using the Total RNA Purification Kit (product #17200) from Norgen Biotek (Ontario, Canada). Novogene (Beijing, China) generated poly-A RNA libraries and performed paired-end RNA sequencing (RNA-seq, read length 150 bp) using a NovaSeq 6000 (Illumina, California, USA). We trimmed sequencing adapters and low quality base calls from the 3' ends of RNA-seq reads using cutadapt26 with parameters -q 20 –minimum-length 36. We aligned reads to the hg19 human genome [63] using STAR [76] with parameters—sjdbOverhang 149—twopassMode Basic—quantMode TranscriptomeSAM—outFilterMultimapNmax 20—alignSJoverhangMin 8—alignSJDBoverhangMin 1—outFilterMismatchNmax 999—outFilterMismatchNover-ReadLmax 0.04—alignIntronMin 20—alignIntronMax 1000000—alignMatesGapMax 1000000. We quantified expression of genes from GENCODE v29 lift37 [77] and corrected for GC bias using salmon44 with parameters–seqBias–gcBias–gencode. We generated RNA-seq quality metrics using the CollectRnaSeqMetrics tool from Picard (https://github.com/broadinstitute/picard). We used PCA to determine which replicates clustered. Within timepoint clusters, we observed additional clustering by batch that we corrected for in downstream analysis.

To identify differentially expressed genes, we imported salmon transcript quantifications and collapsed to the gene level using tximport [78]. We retained 18,299 genes with median DESeq2-normalized count > = 1 across all libraries. We tested for differential gene expression using DESeq2 [72] and classified genes with FDR<5% and LFC>1 as significantly different across pairs of timepoints.

## Gene ontology enrichment of differential genes

We tested if differentially expressed genes were enriched for specific biological processes using the PANTHER statistical overrepresentation test [79] with the GO-Slim Biological Process ontology [75,80]. We ran PANTHER using Fisher's exact test for calculating enrichment and used all 18,299 genes examined in the differential expression analysis as background for the

enrichment tests. We classified ontology terms with fold enrichment$> = 2$ and FDR$<5\%$ as significantly enriched.

## Identification of genes linked to context-dependent peaks

Hi-C: We identified context-dependent peaks that intersect (overlap$> = 1$ base pair) with the "other-end" fragments of "bait-other" Hi-C loops and either end of "bait-bait" loops from previously published adipocyte promoter capture Hi-C data [34,35] using BedTools [66]. We linked peaks to genes that were on the opposite end of the Hi-C "bait-bait" loops. We categorized Hi-C interaction types as "bait-bait" if the "other-end" fragment also covered a bait fragment and "bait-other" if the "other-end" fragment did not cover a bait fragment.

eQTL: We identified context-dependent peaks that overlapped eQTL proxy variants ($r^2>0.8$ with the eQTL lead, 1000G phase 3 European LD calculated using PLINK v1.9[81]) using previously published primary and conditional eQTL mapped in METSIM adipose tissue [4,5] using BedTools [66]. We identified the best eQTL proxy within the peak as the variant with the strongest LD with the lead variant at the signal. If a peak contained proxy variants from both primary and conditional signals with equally strong LD, we selected the primary signal proxy as the best proxy. We also listed all eQTL variants that intersected a peak.

Differential Expression: To provide additional evidence for peak-gene links identified by Hi-C or eQTL, we identified if the linked gene was also differentially expressed (FDR$<5\%$ and LFC$>1$) between any timepoint comparisons. We investigated linking context-dependent peaks to differentially expressed genes based on proximity between the peak and gene TSS, but proximity is indirect and based on the even distribution of peaks from TSS as distance increased, any threshold would have been arbitrary so we concluded that proximity alone was not strong evidence to link a peak and gene (S9 Fig).

## SGBS genotyping and imputation

We genotyped two SGBS DNA samples with 335 samples from a separate study using the Infinium Multi-Ethnic Global array (Illumina, San Diego, CA, USA), which contains over 1.7 million variants. The additional 335 samples were used to calculate genotyping call rates, but all subsequent analyses were performed using only SGBS genotypes. We removed variants with call rate $<95\%$, performed multiple quality checks with the checkVCF.py tool (https://genome.sph.umich.edu/wiki/CheckVCF.py), and oriented alleles relative to the hg19 reference genome [63] using PLINK [81]. We restricted to variants that had the same genotype call in both SGBS samples for downstream analyses. We phased autosomal variants using Eagle v2.4 [82] and imputed missing variants using Minimac3 [83] with the 1000 Genomes (1000G) phase 3 reference panel [84]. The imputation $r^2$ statistic used to assess imputation quality is not meaningful when imputation is performed on a single sample. Therefore, we retained variants with genotype probability (GP) $> 0.9$. In our batch of SGBS cells, a subset of cells showed loss of heterozygosity on regions of chromosomes 7 and 10 (chr7:1–31,000,000 and chr10:131,000,000–135,534,747); variants overlapping these regions were removed prior to downstream analyses.

## ATAC-seq allelic imbalance

To identify heterozygous variants exhibiting allelic imbalance (AI) in SGBS ATAC-seq reads, we first removed reads exhibiting allelic mapping bias and duplicated reads using WASP [85]. We counted reads aligning to each allele of biallelic heterozygous single nucleotide variants using ASEReadCounter [86] with the option–min-base-quality 30 and removed variants that had aligned bases other than the two genotyped alleles. For each SGBS differentiation day, we selected a set of variants to test for AI that had at least 20 total reads combined across both

alleles and at least 3 reads on each allele in greater than 50% of replicates for the given day (3 replicates for D14 and 6 replicates for D0 and D4). We tested for AI separately by day using DESeq2 [72] with the design formula ~0+sample+allele, where 'sample' represents an individual ATAC-seq replicate. Using DESeq2, we tested if the ratio of alternate allele counts to reference allele counts was greater than $\log_2(55/45)$ using a Wald test, estimated dispersions of allelic ratios using maximum likelihood, and adjusted for multiple testing using the BH procedure. We used an LFC threshold of $\log_2(55/45)$ rather than $\log_2(50/50)$, to preferentially select variants showing strong AI, especially given high variability in allelic ratios. We considered variants with FDR<5% to show significant AI.

## Overlap of GWAS signals with context-dependent peaks

We downloaded the NHGRI-EBI GWAS catalog [40] on January 17, 2020 and lifted variant positions from hg38 to hg19 using pyliftover (https://github.com/konstantint/pyliftover), a python implementation of the UCSC liftOver tool [87]; We rescued a subset of variants that did not successfully lift over using variant rsIDs to convert between hg38 and hg19 coordinates. We restricted to significant associations ($p<5\times10^{-8}$) for single nucleotide variants (haplotype associations and variant-variant interactions were removed) that were biallelic in the dbSNP [88] build 151 common variant set. To generate a set of LD-distinct association signals, we performed LD-clumping using swiss (https://github.com/statgen/swiss) in a trait-agnostic manner [5]; the most significant p-value per variant was selected, regardless of trait, and variants within strong LD ($r^2>0.8$, 1000G phase 3 Europeans) and within 1 Mb of another variant with a more significant p-value (not necessarily for the same trait) were removed. However, we retained all variants and associated traits at each signal for reference in supplemental tables. To map GWAS catalog trait terms to standardized ontology terms, we downloaded the GWAS to Experimental Factor Ontology (EFO) mappings file from the GWAS catalog on May 13, 2021 and extracted the EFO term corresponding to each trait. We identified GWAS signals that had at least one proxy variant (LD $r^2>0.8$ with the signal lead variant, 1000G phase 3 Europeans, calculated with PLINK v1.9 [81]) found within context-dependent peaks using BEDTools [66]. For each specific EFO term, we counted the number of signals containing that EFO term, including all variant-trait associations at a signal, not just the strongest association; we only counted each term once per signal. We performed this counting procedure for both the entire LD-clumped GWAS catalog and the subset of the catalog that overlapped the ATAC peak set of interest. Because our goal in using EFO terms is to reduce the complexity of the GWAS catalog traits, we removed any GWAS traits that mapped to 5 or more EFO terms for our analyses that count EFO terms, which only removed <1% of GWAS traits. However, we retained all GWAS traits and EFO terms in S15 Table for reference. To normalize for the overall frequency of an EFO term in the clumped catalog, we divided the number of ATAC counts by the number of total counts for each EFO term and multiplied by 100 to express as a percentage. When ranking by normalized ATAC count to get the top 10 EFO terms for preadipocyte-dependent and adipocyte-dependent peaks, we restricted to terms that had total count $>=$ 100.

## Enrichment of heritability in ATAC peaks

We used stratified LD score regression as implemented in LDSC v1.0.1 [38] to test if ATAC peaks were enriched for heritability of 9 GWAS traits: 8 cardiometabolic traits BMI [89], HDL cholesterol [90], LDL cholesterol [90], triglycerides [90], total cholesterol [90], coronary artery disease [91], WHRadjBMI [89], T2D [92], and rheumatoid arthritis [93] as a negative control. We tested for heritability enrichment separately in 7 different ATAC peak sets: preadipocyte-

dependent peaks, adipocyte-dependent peaks, the top 100,000 consensus peaks for SGBS D0, D4, and D14, and consensus peaks mapped in 17 adipose tissue samples and 11 adipose tissue samples. Using LDSC, we calculated LD scores for ATAC peaks using HapMap3 SNPs [94] and LD calculated from 1000G phase 3 Europeans [84]. We computed partitioned heritability separately for each ATAC peak set using LDSC correcting for the baseline v1.2 model, which consists of 52 genic and functional annotations [38]. We used the regression coefficient z-score reported by LDSC to assess the importance of each ATAC peak set for each trait relative to the baseline model, where a positive z-score means that SNP heritability is increased in a given ATAC peak set relative to the baseline model and a negative z-score means that heritability is decreased in the peak set relative to the baseline [95]. We calculated p-values by testing if the coefficient z-score was greater than 0, assuming a standard normal distribution. We classify results with a p-value threshold of 0.05 as nominally significant and 0.0056 (0.05/9 traits) as significant. We compare the relative importance of each ATAC peak set to heritability for a given trait by comparing coefficient z-scores.

## Prioritization of candidate regulatory elements for functional testing

We identified context-dependent peaks linked to a candidate gene by two or more of our three methods to predict target genes. We identified a further subset of these context-dependent peaks that overlapped a cardiometabolic GWAS signal and an adipose peak. We used further lines of evidence to prioritize these candidate regulatory elements for functional testing including: location of variants closer to the summit of a peak as opposed to the shoulder and literature review of linked gene's relevance to adipose biology

## Transcriptional reporter luciferase assays

SGBS preadipocytes and adipocytes were maintained and transcriptional reporter luciferase assays were performed as previously described [61] with the following changes. Primers were designed to amplify the entire chromatin accessibility region containing variants of interest (S17 Table). Amplified regions containing variant reference and alternate alleles were cloned individually into the XbaI-SbfI restriction sites of the pLS-mP-Luc lentiviral luciferase vectors (a gift from Nadav Ahituv, Addgene plasmid # 106253) or pGL4.23 firefly luciferase reporter vector (Promega) in 'forward' and 'reverse' orientations (named with respect to the genome reference). The variants were cloned upstream of the minimal promoter and verified by Sanger DNA Sequencing. For lentivirus production, HEK293T cells were grown to 70–80% confluency in 100 mm plates and co-transfected with 9.5 µg of a pLS-MP-Luc construct, 8 µg of packaging plasmid (psPAX2, a gift from Didier Trono, Addgene plasmid # 12260), and 2.5 µg of an envelope plasmid (pMD2.G, a gift from Didier Trono, Addgene plasmid # 12259) using Lipofectamine 2000 transfection reagent (Invitrogen). Media was replaced with fresh growth media 18 hours after transfection. Viral supernatant was harvested 48 and 72 hours after transfection and concentrated using 4X Lenti-X concentrator (Clontech). Lentiviral titer was measured using Lenti-X qRT-PCR Titration Kit (Takara Bio), and functional titers were represented as transduction units. For data normalization, empty pLS-MP-Luc and Renilla luciferase vector pLS-SV40-mp-Rluc viruses (a gift from Nadav Ahituv, Addgene plasmid # 106292) were prepared and quantified in a similar manner.

For preadipocytes, 25,000 SGBS cells were plated the day before transduction, and 35,000 SGBS cells were plated and differentiated for adipocytes into 24 well plates and spin-infected with appropriate titer of construct and Renilla virus (S17 Table) in the presence of 10 ug/ml polybrene media. For viral based transcriptional luciferase assays, two independent construct viruses were used for each allele in each orientation and were transduced in tetraplicate wells.

After 8 hrs of transduction, media was replaced with fresh growth media, and luciferase and Renilla activity was measured 48–72 hours post transduction using Dual Luciferase Reporter Assay System (Promega). For plasmid based transcriptional luciferase assays, we used primers (S17 Table) to amplify the regions of interest and we cloned the constructs containing the variants into pGL4.23 firefly luciferase reporter vector (Promega). Five independent clones for each allele in each orientation were cotransfected with Renilla luciferase vector in triplicate wells using Lipofectamine 3000 (Lifetechnologies). Luciferase and Renilla activity were measured after 28hrs of transfection.

For both viral and plasmid based assays, luciferase activity of experimental clones was normalized to Renilla luciferase as well as empty vector activity to control for differences in transfection efficiency. All transcriptional reporter assays were repeated on different days. Data are reported as fold change in activity relative to an empty vector. We used a Student's t-test to compare luciferase activity between alleles and between contexts.

## Supporting information

**S1 Fig. Principal component analysis (PCA) of ATAC-seq read count within peaks for SGBS preadipocytes and adipocytes.** Preadipocytes (D0) are shown in grey, immature adipocytes (D02 and D04) are shown in yellow and red, respectively, and mature adipocytes (D14) are shown in black. Replicates were prepared in four separate batches with batches designated by unique symbols as indicated in the legend. While replicates cluster by batch, PC1 explains 74% of variance and separates preadipocytes from immature and mature adipocytes. Values in S1 Table.
(TIF)

**S2 Fig. UPSET plots of differential chromatin accessibility regions unique or shared between each SGBS differentiation timepoint comparison.** (A-B) Values from S4 Table. (A) Counts of peaks at which earlier timepoints were more accessible than the later timepoints. (B) Counts of peaks at which the later timepoints were more accessible than the earlier timepoints. For both plots, the total number of peaks that show differential accessibility between each pair of timepoints are shown on the bottom right.
(TIF)

**S3 Fig. Comparison of adipose tissue for three subsets of samples.** (A) PCA for PC1 (principal component) vs PC2 for all 17 samples that met quality thresholds. Solid light purple arrows indicate samples that are unique to the 17-sample set (excluded from the 11-sample set). Dashed light purple arrows indicate three previously published samples that have been included in the 11- and 17- sample sets. (B) Hierarchical clustering using the top 10 PCs from PCA. The red dashed line indicates the cutoff used to exclude six samples from the 11-sample set. Dark purple indicates samples in the 11-sample set. Dashed light purple indicates samples in the 3-sample set. Sample numbers correspond to library quality metrics in S10 Table. (C) PCA for PC1 vs PC2 for the 11-sample set. Dashed light purple arrows indicate three previously published samples that have been included. (D) Adipose peak overlap with chromatin states of Roadmap adipose nuclei for the three different sample subsets of adipose consensus peaks using the top 50k peaks for each set. Values in S9 Table. (E) Heatmap of cardiometabolic trait GWAS locus enrichment; rheumatoid arthritis was selected for comparison. Peak sets include two sets of adipose tissue peaks. Values in S12 Table. **, $P < 0.005$; *, $P < 0.05$.
(TIF)

**S4 Fig. PCA of RNA-seq read counts for SGBS preadipocytes and adipocytes.** Preadipocytes (D0) are shown in grey, immature adipocytes (D02 and D04) are shown in yellow and red,

respectively, and mature adipocytes (D14) are shown in black. Replicates were prepared in three separate batches with batches designated by unique symbols as indicated in the legend. While replicates cluster by batch, PC1 explains 54% of variance and separates preadipocytes from immature and mature adipocytes. Values in S8 Table.
(TIF)

**S5 Fig. UPSET plots of differentially expressed genes unique or shared between each SGBS differentiation timepoint comparison.** (A-B) Values from S10 Table. (A) Counts of genes that were more highly expressed in an earlier timepoint than a later timepoints. (B) Counts of genes that were more highly expressed in a later timepoints than an earlier timepoints. For both plots, the total number of genes that were differentially expressed between each pair of timepoints are shown on the bottom right.
(TIF)

**S6 Fig. Allelic imbalance of variant in context-dependent peak.** Allele counts of variant rs11039149 within context-dependent peak, peak23801, that is more accessible in D4 compared to D0. This peak was linked to *NR1H3* through two methods. Full allelic imbalance results including counts at this variant in S15 Table.
(TIF)

**S7 Fig. Replicate experiment of *SCD* transcriptional activity.** (A-B) Values in S18 Table. (A) Replicate of Fig 4 using the same viral vectors transduced on an independent day. A 592-bp genomic region surrounding peak19405 containing the rs603424-G allele shows increased transcriptional activity compared to the rs603424-A allele in the forward (FWD) and modestly-increased transcriptional activity in the reverse (RVS) orientations in adipocytes (tested at day 7), the context in which chromatin was more accessible compared to preadipocytes. The rs603424-G allele also shows increased transcriptional activity compared to the rs603424-A allele in the RVS orientations in preadipocytes. Dots represent two independent constructs assayed in four replicates each. Luciferase activity was normalized relative to an empty vector (EV). (B) Replicate experiment using plasmid transfections. A 481-bp genomic region surrounding peak19405 containing the rs603424-G allele shows increased transcriptional activity compared to the rs603424-A allele in the forward and reverse orientations in adipocytes (tested at day 3.5), the context in which chromatin was more accessible compared to preadipocytes. Dots represent 5 independent constructs assayed in three replicates each.
(TIF)

**S8 Fig. Replicate experiments of *EYA2* transcriptional activity.** (A-D) Values in S18 Table. (A-B) Replicates of Fig 5 using the same viral vectors transduced on an independent day. (A) A 419-bp genomic region surrounding peak81750 containing the rs555966194-C allele showed increased transcriptional activity compared to the rs555966194-G allele in the reverse (RVS) orientation, but not the forward (FWD), in adipocytes (tested at day 10) and preadipocytes, although transcriptional activity was higher in adipocytes, the context in which chromatin was more accessible compared to preadipocytes. Dots represent two independent constructs assayed from four replicates each. Luciferase activity was normalized relative to an empty vector (EV). (B) A 288-bp genomic region containing the rs59791349-C allele shows increased transcriptional activity compared to the rs59791349-T allele in the reverse orientation and in both preadipocytes and adipocytes (tested at day 9). Dots represent two independent constructs assayed from four replicates each. (C-D) Replicate experiments using plasmid transfections. (C) A 419-bp genomic region surrounding peak81750 containing the rs555966194-C allele shows increased transcriptional activity compared to the rs555966194-G allele in the forward orientation, but not the reverse, in adipocytes (tested at day 3.5). Dots represent 5

independent constructs assayed in three replicates each. (D) A 312-bp genomic region containing the rs59791349-C allele shows increased transcriptional activity compared to the rs59791349-T allele in the forward and reverse orientation and in adipocytes (tested at day 3.5). Dots represent 5 independent constructs assayed in three replicates each.
(TIF)

**S9 Fig. Distances from the transcription start site for differentially accessible peaks linked to differentially expressed genes.** Histogram of distances of differentially accessible peaks to a TSS for all differentially expressed genes within 500,000 bases (n = 74,058).
(TIF)

**S1 Table. ATAC-seq library metrics for SGBS libraries.** ATAC-seq libraries of SGBS preadipocytes (D00), immature adipocytes (D02: not included in final analyses, and D04), and adipocytes (D14) with batch, sequencing and alignment metrics.
(XLSX)

**S2 Table. Summary of chromatin accessibility consensus peaks and genes for SGBS differentiation timepoints.** Consensus peaks were defined as the union of chromatin accessibility region accessible in majority of replicates for timepoint, overlapping by 1 or more base pairs and consensus genes were defined as those expressed in majority of replicates for timepoint (see Methods for more information).
(XLSX)

**S3 Table. Summary of context-dependent peaks and genes.** Total context-dependent peaks and genes (DESeq2, LFC>1, FDR<5%) identified for each timepoint comparison.
(XLSX)

**S4 Table. Context-dependent peaks for all differentiation timepoints.** Context-dependent peaks (DESeq2, LFC>1, FDR<5%) and which differentiation timepoints comparisons they were more accessible in with Log Fold Change (LFC), P-value, and Q-value reported. Visualization of top 10k context-dependent peaks in Fig 1B. Visualization of overlap of context-dependent peaks between timepoints in S2 Fig.
(XLSX)

**S5 Table. Gene ontology for context-dependent peaks.** Gene ontology results from GREAT for top 10k context-dependent peaks for each differential timepoint comparison. See Methods for GREAT parameters.
(XLSX)

**S6 Table. Transcription Factor motifs enriched in context-dependent peaks.** Transcription factor enrichment results from HOMER for preadipocyte-dependent (LFC>1 in D0>D4 and D0>D14, Q<0.05 in D0>D4 or D0>D14) or adipocyte-dependent (LFC>1 in D4>D0 and D14>D0, Q<0.05 in D4>D0 or D14>D0) peaks. See Methods for HOMER parameters.
(XLSX)

**S7 Table. ATAC-seq library metrics for adipose tissue libraries.** Adipose tissue ATAC-seq library metrics with batch, sequencing and alignment metrics.
(XLSX)

**S8 Table. Peak Overlap With Chromatin States.** Percent peak overlap with chromatin states of Roadmap Epigenomics Project adipose nuclei for three sets of adipose consensus peaks and preadipocyte- and adipocyte-dependent peaks. Visualization of peak overlap in Fig 1D and 1E.
(XLSX)

**S9 Table. RNA-seq library metrics for SGBS libraries.** RNA-seq libraries of SGBS preadipocytes (D00), immature adipocytes (D02: not included in final analyses, and D04), and adipocytes (D14) with batch, sequencing and alignment metrics. All libraries sequenced with a read length of 150 bases.
(XLSX)

**S10 Table. Context-dependent genes for all differentiation timepoints.** List of context-dependent genes (DESeq2, LFC>1, FDR<5%) and which timepoints they were more expressed in with Log Fold Change (LFC), P-value, and Q-value reported. Visualization of context-dependent genes in Fig 1C. Visualization of overlap of context-dependent peaks between timepoints in S5 Fig.
(XLSX)

**S11 Table. Gene ontology for context-dependent genes.** Gene ontology results for differentially expressed genes between timepoints using PANTHER. See Methods for PANTHER parameters.
(XLSX)

**S12 Table. Context-dependent peaks linked to genes by any of three approaches.** We identified context-dependent peaks linked to genes by any of three methods (Hi-C overlap, eQTL overlap, or Hi-C/eQTL overlap where the gene was differentially expressed) as shown in Fig 2. Peaks linked to genes that overlapped a GWAS variant, an adipose tissue consensus peak, or demonstrated allelic imbalance are also indicated.
(XLSX)

**S13 Table. GWAS enrichment in indicated peak sets for select traits.** GWAS enrichment results as shown in Figs 3A and S3E for indicated peaks sets in select traits.
(XLSX)

**S14 Table. GWAS specific ontology (EFO) terms for GWAS signals overlapping context-dependent peaks.** GWAS traits that mapped to 5 or more EFO terms were removed. "Total Count of Signals From Catalog" is the total number of LD-clumped GWAS signals that mapped to a given EFO term in the full clumped GWAS catalog before overlap with ATAC peaks. We only retained EFO terms with "Total Count of Signals From Catalog" of at least 100. The normalized count is the count overlapping the specified ATAC peaks divided by the total count from the GWAS catalog multiplied by 100. The top 10 results ranked by preadipocyte- and adipocyte-dependent normalized counts are shown in Fig 3B.
(XLSX)

**S15 Table. GWAS variants overlapping context-dependent peaks.** All GWAS variants ($r2>0.8$) overlapping context-dependent peaks, not limited to peaks linked to genes. Due to rounding of p-values in the GWAS catalog to one digit, some signal lead variants have multiple traits that share the most significant p-value. In such cases, multiple best traits for the same lead variant are listed in separate rows of the table. We included all listed GWAS traits and Experimental Factor Ontology (EFO) terms for a signal, regardless of how many EFO terms a trait mapped to. This is in contrast to the enrichment analysis in S14 Table where traits that mapped to 5 or more EFO terms were removed.
(XLSX)

**S16 Table. Allelic Imbalance.** Target Gene is identified if linked by two or more approaches. A plot of allelic imbalance at the NR1H3 locus is shown in S6 Fig.
(XLSX)

**S17 Table. Experiment design for luciferase assays.** Experimental design for luciferase assays including primer sequences, chromosome positions, and experimental parameters.
(XLSX)

**S18 Table. Luciferase Results.** Transcriptional activity measured as luciferase activity normalized to empty vector (EV) for all luciferase reporter assays performed. Visualization of results in Figs 4, 5, S7, and S8.
(XLSX)

## Acknowledgments

We thank Kristina Garske and Paivi Pajukanta for assistance with Hi-C data, Jeremy Simon for interpretation, Jason Luo and personnel of the University of North Carolina Mammalian Genotyping Core for genotype data, and the participants in the METSIM study who donated adipose tissue samples. We acknowledge ENCODE and Michael Snyder's lab (Stanford University) for ATAC-seq data and the NIH Roadmap Epigenomics Consortia for chromatin state data.

## Author Contributions

**Conceptualization:** Hannah J. Perrin, Kevin W. Currin, Karen L. Mohlke.

**Data curation:** Hannah J. Perrin, Kevin W. Currin, Markku Laakso.

**Formal analysis:** Hannah J. Perrin, Kevin W. Currin.

**Funding acquisition:** Hannah J. Perrin, Kevin W. Currin, Karen L. Mohlke.

**Investigation:** Hannah J. Perrin, Kevin W. Currin, Swarooparani Vadlamudi, Gautam K. Pandey, Kenneth K. Ng.

**Resources:** Martin Wabitsch, Markku Laakso, Karen L. Mohlke.

**Supervision:** Michael I. Love, Karen L. Mohlke.

**Validation:** Hannah J. Perrin, Kevin W. Currin, Swarooparani Vadlamudi, Gautam K. Pandey, Kenneth K. Ng.

**Visualization:** Hannah J. Perrin, Kevin W. Currin.

**Writing – original draft:** Hannah J. Perrin, Kevin W. Currin, Karen L. Mohlke.

**Writing – review & editing:** Hannah J. Perrin, Kevin W. Currin, Swarooparani Vadlamudi, Gautam K. Pandey, Kenneth K. Ng, Martin Wabitsch, Markku Laakso, Michael I. Love, Karen L. Mohlke.

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
