## [Decision Letter · Decision Letter 0]

29 Jul 2021

Dear Dr Mohlke,

Thank you very much for submitting your Research Article entitled 'Chromatin accessibility and gene expression during adipocyte differentiation identify context-dependent effects at cardiometabolic GWAS loci' to PLOS Genetics.

The manuscript was fully evaluated at the editorial level and by independent peer reviewers. The reviewers appreciated the attention to an important problem, but raised some substantial concerns about the current manuscript. Based on the reviews, we will not be able to accept this version of the manuscript, but we would be willing to review a much-revised version. We cannot, of course, promise publication at that time.

If you decide to revise the manuscript for further consideration at PLOS Genetics, please aim to resubmit within the next 60 days, unless it will take extra time to address the concerns of the reviewers, in which case we would appreciate an expected resubmission date by email to plosgenetics@plos.org.

[LINK]

We are sorry that we cannot be more positive about your manuscript at this stage. Please do not hesitate to contact us if you have any concerns or questions.

Yours sincerely,

Chris Cotsapas, PhD

Associate Editor

PLOS Genetics

Wendy Bickmore

Section Editor: Epigenetics

PLOS Genetics

Reviewer's Responses to Questions

**Comments to the Authors:**

Reviewer #1: In this submission, Perrin et al. aim to identify mechanisms downstream of cardiometabolic GWAS loci through transcriptional and epigenetic profiling of a human cell model of preadipocytes and adipocytes. The relative paucity of eQTLs at GWAS loci may relate partly to the context (cell type, stimulus conditions) of eQTL studies. Like other disease-relevant tissues, adipose tissue is heterogeneous; GWAS variants may therefore act specifically in adipocytes, preadipocytes, or other constituent cells. Using their cell line, the authors identify peaks that are accessible at specific developmental stages, and subsequently assess these for disease relevance.

The authors begin by profiling chromatin accessibility in SGBS cells (Simpson-Golabi-Behmel Syndrome, a preadipocyte cell line) with ATAC-seq at D0 (preadipocyte), D4 (immature adipocyte) and D14 (mature adipocyte) of differentiation. They identify consensus peaks by overlap at each developmental stage. Differentially accessible ("context-dependent") peaks are identified between each pair of timepoints. Most changes in chromatin accessibility appear to occur between day 0 and day 4, then persist to day 14.

ATAC-seq peaks that are more accessible in preadipocytes (D0>D4 and D0>D14) are identified, along with a similar set for adipocytes. The paper defines the adipocyte set as D4>D0 and D14>D0, but the D14>D4 and D14>D0 set would seem to be more specific for mature adipocytes; further motivation for this choice, and examination of its effect on their overall results, would be helpful. Both preadipocyte- and adipocyte-dependent peaks were enriched near genes involved in relevant metabolic processes and appropriate transcription factor motifs, suggesting they capture relevant biology.

In parallel, the authors generate a consensus map of chromatin accessibility in subcutaneous adipose tissue from 11 healthy donors (17 prior to QC) using similar methods. Sequencing details are somewhat heterogeneous across the study, with variable read length and sequencing method (paired-end vs. single-end). This is addressed to an extent in their QC but remains a potential limitation.

To link changes in chromatin accessibility to potential gene regulatory changes, the authors use RNA-seq to identify gene expression alterations at each stage of adipocyte differentiation. They link context-dependent peaks with potential gene targets by consensus of 2/3 methods: (a) overlap with promoter-capture Hi-C regions, (b) overlap with adipose eQTL signals, and (c) Hi-C overlap or eQTL overlap with additional context-dependent expression. Each of these methods has significant limitations in isolation. eQTL signals overlapping context-dependent accessibility regions do not necessarily have a context-dependent effect. Hi-C regions are frequently large, and the authors included overlaps as small as a single base pair. A more stringent overlap threshold may more precisely identify candidate peaks of interest.

A total of 5,145 peaks were linked to the same gene by >=2 approaches, with 78 peaks linked to the same gene by all three approaches. The 78 peaks that link to the same gene by all three methods would seem logical candidates for further mechanistic exploration, but the authors focus on the larger set of traits linked by >=2 methods. They identify GWAS signals (proxies with r^2>0.8) that overlap context-dependent peaks and link these to gene targets using their consensus method. While a number of these may represent disease mechanisms, these should be viewed as mechanistic candidates. Simple overlap is not sufficient to establish causal mechanistic links between GWAS variants, putative context-dependent regulatory elements within context-dependent chromatin, and downstream genes. Further work will be required to validate hypotheses that arise from this work.

Supporting the relevance of context-dependent chromatin regions, stratified LD score regression is used to demonstrate heritability enrichment for waist:hip ratio adjusted for BMI in adipocyte-dependent and adipose tissue peaks but not in preadipocyte-dependent peaks. Coronary artery disease was also enriched in adipose tissue peaks and SGBS peaks defined at day 0 and day 4. Allelelic imbalance is observed in several peaks, but this analysis was limited by the fact that all SGBS cells have the same genotype.

Data from this study will be a useful resource for future studies of GWAS mechanism within adipose tissue, but at this point specific hypotheses are not validated. Demonstration of context-dependent chromatin accessibility is an important first step, but the present study cannot identify specific regulatory elements that are contained within these peaks. The SGBS cell line offers an elegant approach to dissociate preadipocyte and adipocyte biology, but disease relevance may be somewhat limited by the artificial conditions in which the cell line is grown.

Reviewer #2: Perrin et al present a nice manuscript regarding ATAC seq and RNA seq in SGBS cells, day 4 and 14

compared to ATAC seq from 11 SAT human samples (METSIM). Found > 50 K context dependent chromatic accessibility in all timepoint comparisons. Linked context dependent peaks based on Hi-C data overlap with eQTL and gene expression. Found GWAS for cardiometabolic traits, adipocyte dependent peaks enriched for WHR.

Overall the goals are good: Find out how GWAS loci work, Map regulatory elements with Chromatic accessibility (ATAC Seq) and in an appropriate cell/tissue (Adipose tissue important in CM traits, lots of eQTLs in adipose). This group has a track record in this area and the methods seem sound.

I think a few things will improve the paper.

-16K context dependent peaks linked to a gene -> 5K linked to 1.6K genes (Fig 2 E) -> 78 linked to same gene through all 3 approaches...I wonder what are these 78 genes? Seems like these are low hanging fruit as it were?

-p.10…Applaud the use of many different ways to try to link peaks to genes…But seems like this is necessarily limited by the availability of Hi-C data from preadipocytes. Was this not looking “under the lamp post where there is light”? Similar to having eQTL data from adipocytes but not pre-adipocytes. I think even more focus could potentially be given to adipocytes given this limitation.

-Nice to see the heritability enrichment (p.12 and Table S13). The striking findings with WHR are important as are the findings related to HDL, TG, and CAD which all point to a “metabolic syndrome” overlap. Overall the data presented on table S13 should probably emphasize normalized count overlapping adipose dependent peaks rather than any context dependent peak. At least when I order the table in this regard it makes a lot of biological sense. Associations with WHR, HDL, birth weight, TG, CVD, bone density Afib are all expected. However, other interesting biological insight is that breast cancer is in the top 25 as are eczema, and many pulmonary traits (FEV1). This is quite interesting. Would be interested to see if genetically programmed WHR is causally related to FEV1 as a later paper. Also the relationship to intraocular pressure is not immediately intuitive though I am not an eye or glaucoma expert. That table also seems a bit strange to me in one way. There are about 290 traits that have the same number of “total count of signals from catalog, n=1277 with same number of “count overlapping any context dependent peak, n=233) giving identical “normalized count overlapping…, n=18.25). Why is this? I guess I would like to see a bit more of the discussion focused on medical relevance and the results less about presenting "numbers" of peaks with overlap (which could be gleaned from the figures)...More about why all this work is medically relevant would help the reader.

Minor

-I understand looking at enrichment of transcription factor motifs (p.6-7) in ATAC seq peaks…But less so the pathway analyses. Pathway analyses focused on the “nearest” gene (Table S5) will often be misleading as often times the “nearest” gene is not relevant (as elegantly shown in this and other papers).

-"Context dependent" is a reasonable phrase but non-specific. Could “differentiation-specific” be used?

-Seems like there has been a similar ATAC Seq study in more commonly used 3T3-L1 cells (Discussion) which may be mentioned in the Intro. What else was the overlap in findings from the mouse to the human cell line? We all want to understand if the mouse cell line can be used as it is a bit easier than SGBS (which have a limited supply).

-Fig 1a indicates ATAC seq in 17 people but abstract says 11 people were used, I see in the Figure legend a subset of 11 was used but what happened to the other 6 (33% of the samples)? p.7 indicates they were outliers due to PCA. But why even bother mentioning these or including the data. You could simply omit all references to the 6 samples from all analyses by saying they did not meet QC or other standards.

-Fig 1b…order is D0, D14, D4 and 1c is D0, D4, D14…why the difference? Shouldn’t order be temporal?

**Have all data underlying the figures and results presented in the manuscript been provided?**

Reviewer #1: None

Reviewer #2: Yes

PLOS authors have the option to publish the peer review history of their article (what does this mean?). If published, this will include your full peer review and any attached files.

Reviewer #1: No

Reviewer #2: No

---

## [Decision Letter · Decision Letter 1]

7 Oct 2021

Dear Dr Mohlke,

We are pleased to inform you that your manuscript entitled "Chromatin accessibility and gene expression during adipocyte differentiation identify context-dependent effects at cardiometabolic GWAS loci" has been editorially accepted for publication in PLOS Genetics. Congratulations!

Yours sincerely,

Chris Cotsapas, PhD

Associate Editor

PLOS Genetics

Wendy Bickmore

Section Editor: Epigenetics

PLOS Genetics

Comments from the reviewers (if applicable):

Reviewer's Responses to Questions

**Comments to the Authors:**

Reviewer #1: Perrin et al. revise their submission examining chromatin accessibility and gene expression in the SGBS model of adipocytes and preadipocytes. They have addressed several concerns raised in initial review and the paper now reads more clearly, with improved justification.

The authors continue to define "adipocyte-dependent" peaks as D4>D0 and D14>D0. While this is less specific for mature adipocytes than the D14>D4 and D14>D0 peak set, it is clarified that there were only 233 peaks specific to mature adipocytes. As chromatin accessibility appears to remain relatively stable between D4 and D14, combining immature and mature adipocytes into a single category seems reasonable. Similarly, the decision to focus on peaks that are linked to genes by 2/3 approaches is better motivated.

The revised manuscript more clearly acknowledges a few important limitations. Sequencing methods varied somewhat throughout the study, representing a source of technical variation. It is reassuring that clustering of samples did not appear to be significantly affected. They more clearly acknowledge the challenge of translating overlaps with GWAS loci into molecular mechanism and appropriately call for further studies.

This study will be an important resource for studies of GWAS mechanisms in adipose tissue. Potential dissociation of preadipocyte and adipocyte biology is particularly attractive, though the usual caveats pertaining to cell lines continue to apply.

Reviewer #2: my comments have been addressed

**Have all data underlying the figures and results presented in the manuscript been provided?**

Reviewer #1: None

Reviewer #2: Yes

PLOS authors have the option to publish the peer review history of their article (what does this mean?). If published, this will include your full peer review and any attached files.

Reviewer #1: No

Reviewer #2: No

**Data Deposition**

http://datadryad.org/submit?journalID=pgenetics&manu=PGENETICS-D-21-00875R1

**Press Queries**

---

## [Editor Report · Acceptance letter]

22 Oct 2021

PGENETICS-D-21-00875R1 

Chromatin accessibility and gene expression during adipocyte differentiation identify context-dependent effects at cardiometabolic GWAS loci 

Dear Dr Mohlke, 

We are pleased to inform you that your manuscript entitled "Chromatin accessibility and gene expression during adipocyte differentiation identify context-dependent effects at cardiometabolic GWAS loci" has been formally accepted for publication in PLOS Genetics! Your manuscript is now with our production department and you will be notified of the publication date in due course.

With kind regards,

Katalin Szabo

PLOS Genetics

On behalf of:
